# Myosin-binding protein C regulates the sarcomere lattice and stabilizes the OFF states of myosin heads

Anthony L. Hessel [1,2] ✉, Nichlas M. Engels[3], Michel N. Kuehn[1,2], Devin Nissen[4], Rachel L. Sadler [5], Weikang Ma [4], Thomas C. Irving[4], Wolfgang A. Linke [1] & Samantha P. Harris [5] ✉

Muscle contraction is produced via the interaction of myofilaments and is regulated so that muscle performance matches demand. Myosin-binding protein C (MyBP-C) is a long and flexible protein that is tightly bound to the thick filament at its C-terminal end (MyBP-C[C8C10]), but may be loosely bound at its middle- and N-terminal end (MyBP-C[C1C7]) to myosin heads and/or the thin filament. MyBP-C is thought to control muscle contraction via the regulation of myosin motors, as mutations lead to debilitating disease. We use a combination of mechanics and small-angle X-ray diffraction to study the immediate and selective removal of the MyBP-C[C1C7] domains of fast MyBP-C in permeabilized skeletal muscle. We show that cleavage leads to alterations in crossbridge kinetics and passive structural signatures of myofilaments that are indicative of a shift of myosin heads towards the ON state, highlighting the importance of MyBP-C[C1C7] to myofilament force production and regulation.

Muscle sarcomeres are composed of an interdigitating array of myosin-containing thick and actin-containing thin filaments (Fig. 1a). Contraction force is generated via a carefully orchestrated action where myosin-based motor proteins of the thick filaments interact with the actin molecules of the thin filaments to generate force and length changes by a process called crossbridge cycling[1,2]. Myosin molecules are hetero-multimers and have a tail, neck (regulatory and essential light chain region), and head (motor) regions, which are packed into the thick filament as a three-stranded quasi-helical array forming repeating "crowns" of three sets of myosin heads separated by ~120° from each other[3]. As one moves along the thick filament axially, crowns arise at ~14.3 nm intervals, each rotated by 40°, so that the crown orientation repeats every three turns / ~ 43 nm (Fig. 1b). Within this repeat, each of the three myosin crowns has slightly different features and are denoted here as $Cr_1$, $Cr_2$, and $Cr_3$. Thick filament crowns are also classified as occurring in 3 distinct segments of the thick filament, with 6 crowns in the P- (proximal), 27 crowns in the C- (central), and 18 crowns in the D- (distal) zone, where C-zone crowns associate with an extra protein called myosin-binding protein C (MyBP-C) with a stoichiometry of ~three MyBP-C molecules spaced at 43 nm intervals (Fig. 1a-b)[4–7].

In addition to the textbook description of a thin filament-based regulation scheme of muscle contraction[8,9], there is a growing appreciation of thick filament-based regulation mechanisms acting in parallel[10,11]. Structurally, a proportion of myosin heads are in an OFF conformation, unable to interact with actin, and docked close to the thick filament backbone in quasi-helical tracts. Other myosin heads are in an ON state positioned away from the thick filament backbone, where they can readily bind to the thin filaments during contraction (Fig. 1c)[10,12]. In resting muscle, most heads are in the OFF state with only a few in the ON state. The strain-dependent thick filament activation model[13] posits that these few ON state heads generate strain in the thick filament early in activation that causes a cooperative OFF-to-ON transition of myosin heads to participate in contraction. Thick filament activation mechanisms are also likely to be involved in myofilament length-dependent activation (LDA), the phenomenon whereby the sarcomeres can generate more force at longer sarcomere lengths (SL)

[1]Institute of Physiology II, University of Muenster, Muenster, Germany. [2]Accelerated Muscle Biotechnologies Consultants, Boston, MA, USA. [3]Department of Cellular and Molecular Medicine, University of Arizona, Tucson, AZ, USA. [4]BioCAT, Department of Biology, Illinois Institute of Technology, Chicago, IL, USA. [5]Department of Physiology, University of Arizona, Tucson, AZ, USA. ✉e-mail: anthony.hessel@uni-muenster.de; samharris@arizona.edu

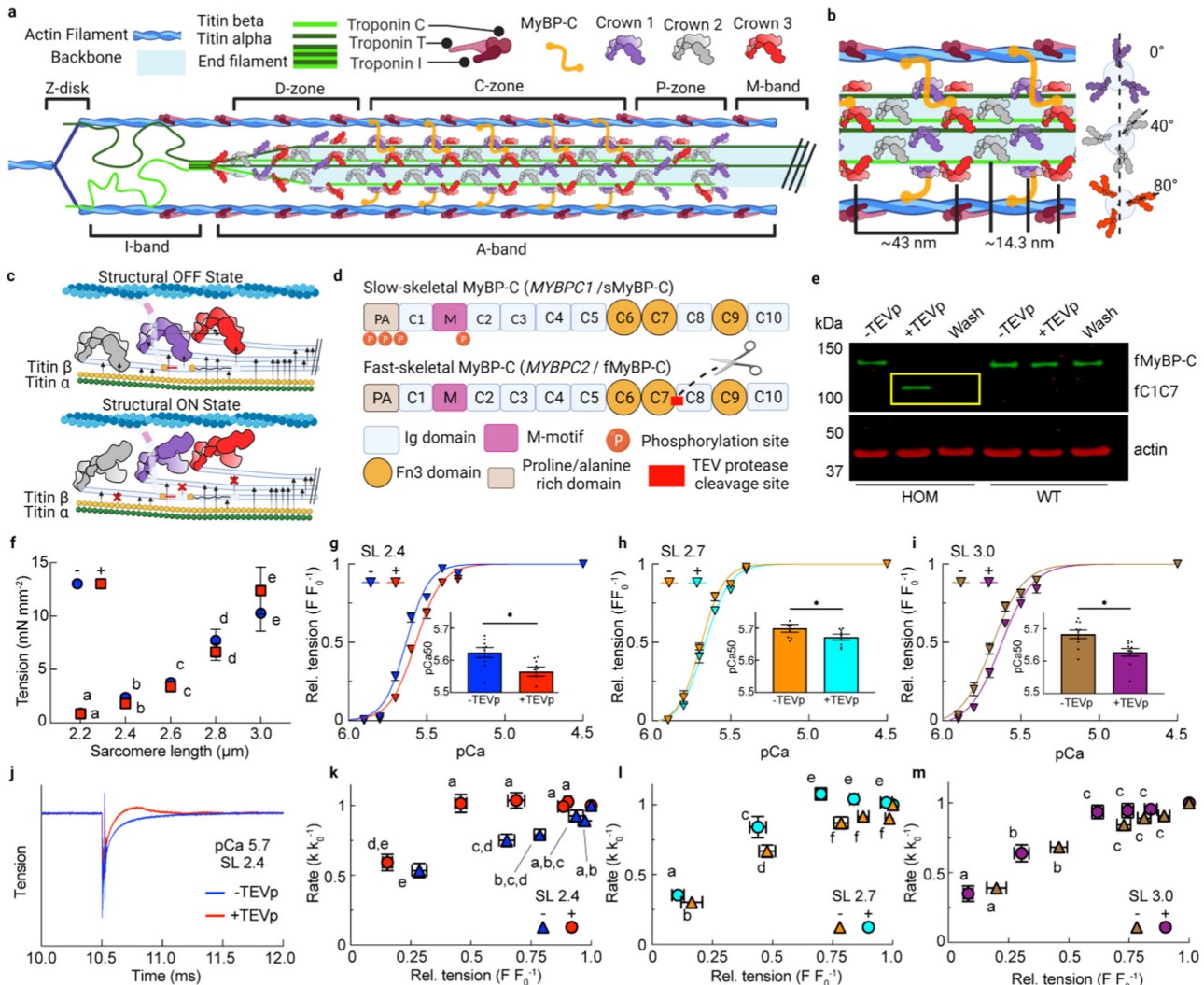

**Fig. 1 | Study of MyBP-C in skeletal muscle by targeted, acute and specific cleavage in situ. a** Schematic of a skeletal half-sarcomere. **b** Expanded view of crown placement on thick filaments. **c** Schematic of one ~43 nm repeat in the C-zone, with crowns shown in the structural OFF and ON states (details in text). Arrows indicate the suggested interactions between different elements[29,31,51]. **d** Domain layout of fast and slow twitch MyBP-C, with the TEV protease cleavage site of SNOOPC2 mouse line indicated. **e** Western blot of tagged fast MyBP-C isoform (fMyBP-C) from homozygous and wildtype SNOOPC2 psoas, before and after fast isoform cleavage (fMyBP-C$^{\mathrm{C1C7}}$−cleaved N-terminal domains). The experiment was repeated 3 times for the HOM condition and 1 time for WT. **f**−**m** Mechanical measurements of permeabilized psoas fibers from SNOOPC2 muscle, before (−) and after (+) TEV protease treatment (N-terminal domains removed) for passive

tension-SL (**f**), relative tension-pCa curves at 2.4 (**g**), 2.7 (**h**), and 3.0 μm (**i**) SL, representative tension $k_{\mathrm{tr}}$ curves before and after TEV protease treatment, (**j**) and normalized $k_{\mathrm{tr}}$ vs. relative tension at 2.4 (**k**), 2.7 (**l**), and 3.0 μm (**m**) SL. Statistics throughout are repeated-measures ANOVA designs followed by a Tukey honestly significant difference (HSD) post-hoc test on statistically significant main effects. *$P < 0.05$ after treatment at each SL. Stats in (**k**, **f**−**m**) are reported via connecting letters, where conditions with different letters are significantly different. Data throughout was reported as mean ± SE. Experimental dataset derived from 49 fiber bundles from psoas muscles of 10 SNOOPC2 mice (6 female/4 male). Further descriptive and statistical details are in Supplementary Tables 1–8. Created with BioRender.com.

at a given level of activating calcium[14]. LDA is now understood as a general feature of striated muscles and is often found to be dysregulated in skeletal- and cardiomyopathies[10,14,15].

LDA is thought to involve two major players: the large titin proteins that run along the half-thick filament length in the A-band and bridge the thin and thick filaments in the I-band of the sarcomere[16,17], and the thick filament-bound MyBP-C[4]. However, these molecules could also be involved more generally in mechanisms of thick-filament activation, where the activation sensitivity of cross-bridges is tunable[4,16,18]. MyBP-C has three paralogs that are encoded by distinct genes that are generally expressed by muscle type: cardiac (encoded by *MYBPC3*), slow skeletal (sMyBP-C, encoded by *MYBPC1*), and fast skeletal (fMyBP-C, encoded by *MYBPC2*) (Fig. 1d). Mutations in the skeletal muscle paralogs lead to debilitating myopathies in humans

including severe and lethal forms of distal arthrogryposis myopathy[19,20] and lethal congenital contracture syndrome[21], suggesting a critical role in healthy sarcomere function. In addition, fMyBP-C knockout mice present altered force generation with myosin heads shifted towards the ON state[21,22], suggesting that MyBP-C helps stabilize the myosin OFF state and/or suppresses OFF-to-ON transitions[22–24]. Importantly, MyBP-C effects can be blunted via post-translational modifications such as phosphorylation[25–27], which can be altered in some myopathies but also in healthy muscle during exercise, e.g., via increased PKA/PKC activity[28]. The mechanism(s) by which MyBP-C affects the myosin ON/OFF state is not clear, but the phenomenon is remarkable because there are relatively few MyBP-C molecules (up to ~54 covering 9 crowns per half-thick filament) to regulate the behavior of up to ~300 myosin heads per thick filament.

To evaluate the structural and functional effects of MyBP-C in sarcomeres, we developed the SNOOPC2 mouse line (Fig. 1d, Supplementary Fig. 1a), which has an engineered tobacco etch virus protease (TEV$_P$) recognition site inserted between the C7 and C8 domains of the *MYBPC2* gene. This addition allows for immediate, controllable, and specific cleavage of the N-terminal region of fMyBP-C (from the N-terminal proline/alanine sequence through the C7 domain; MyBP-C$^{C1C7}$) in permeabilized skeletal muscle, leaving only the C-terminal portion (C8–C10) anchored to the thick filament. The short SnoopTag insertion occurs upstream of C8 and so should not disrupt the proposed OFF-state stabilizing role of the Cr1 and Cr3 heads, as recently described[29–31]. This powerful experimental approach allows for the study of changes that occur before and after the removal of MyBP-C$^{C1C7}$ within the same preparation, removing variation caused by using different samples between conditions and improving statistical power.

In this work, we report that after cleavage, the thin filaments are shorter, a result consistent with an intimate interaction of MyBP-C with thin filaments as indicated in recent cryo-electron tomography studies[29,30]. $Ca^{2+}$ sensitivity is reduced at shorter sarcomere lengths, and cross-bridge kinetics are increased across sarcomere lengths at submaximal activation levels, demonstrating a role in cross-bridge kinetics. X-ray structural signatures of the thick filaments suggest that cleavage also shifts myosin heads towards the ON state—a marker that could be accounted for via increased cross-bridge kinetics at submaximal $Ca^{2+}$ and/or a change in the force transmission pathway. Taken together, we conclude that MyBP-C$^{C1C7}$ domains play an important role in contractile performance.

## Results and discussion
### fMyBP-C$^{C1C7}$ cleavage alters contraction properties

We used psoas muscle from homozygous SNOOPC2 mice for this evaluation because nearly all fibers are of fast-twitch composition[32]. Western blots confirmed successful cleavage of fMyBP-C in permeabilized homozygous SNOOPC2 psoas (Fig. 1e) while not targeting sMyBP-C (Supplementary Fig. 1b) or wildtype fMyBP-C (Fig. 1e). Furthermore, passive tension of fMyBP-C was not affected by TEV$_P$ treatment in fibers from homozygous or wildtype mice in absolute or relative terms (Fig. 1f; Supplementary Fig. 1c). We next evaluated the tension-pCa ($-\log[Ca^{2+}]$) relationship in permeabilized fiber bundles before and after cleavage of N-terminal domains of fMyBP-C at shorter (2.4 μm), middle (2.7 μm), and longer (3.0 μm) SLs (Fig. 1g–i; Supplementary Fig. 1c). However, due to compliance resulting from attaching the cell with silicone rubber, at maximal activation the SL shortened to 2.1 μm (shorter), 2.4 μm (middle), and 2.7 μm (longer). Cleavage caused a statistically significant rightward shift of the tension–pCa relationship, as quantified by the pCa$_{50}$ (i.e., the pCa at which active force was half-maximal) at all SLs (Fig. 1g–i). Although pCa$_{50}$ was altered, LDA was still observed from short to middle and long SLs (i.e., increasing pCa$_{50}$) in samples both before and after MyBP-C$^{C1C7}$ removal (overlaid in Supplementary Fig. 1d, e). While cleavage impacted submaximal activation levels, it had no statistically significant effect on absolute maximum tension across SLs (Supplementary Fig. 1d, e). Next, we assessed the rates of force redevelopment following a slack/restretch maneuver ($k_{tr}$), a measure of cross-bridge cycling, across a range of calcium concentrations (Fig. 1j–m). At 2.4 and 2.7 μm SL, the $k_{tr}$ vs. force relationships were similar to previous studies[23,33], with $k_{tr}$ increasing with relative force (Fig. 1k, l). Qualitatively, cleaved samples produced a consistent overshoot of the tension near pCa$_{50}$ before settling to expected values (example in Fig. 1j), which has been reported previously in fast twitch and slow twitch fibers[34]. At both short and middle SLs, the loss of fMyBP-C$^{C1C7}$ increased $k_{tr}$ at intermediate but not low or maximal active force levels, while $k_{tr}$ was not different at the longer SL (Fig. 1k–m). Therefore, $k_{tr}$ was less affected by fMyBP-C cleavage at longer vs. shorter SLs. These data are indicative of

faster rates of myosin cross-bridge cycling after cleavage of fMyBP-C at intermediate calcium activation, consistent with effects observed following loss of cardiac MyBP-C$^{C0C7}$ in cardiomyocytes and in cardiac and fast skeletal knockout mice[23,33]. Detailed statistical information for mechanical datasets is provided in Supplementary Tables 1–8. Taken together, these data suggest that fMyBP-C$^{C1C7}$ modulates force generation and crossbridge cycling kinetics, specifically at moderate levels of activation (~40–80% of maximal force), in agreement with previous observations in a cardiac MyBP-C TEVp cleavage model, knockout mouse models, and in vitro biochemical studies[4,19,21,23,33,35].

### fMyBP-C$^{C1C7}$ cleavage alters the myofilament lattice

To determine structural effects in sarcomere proteins due to fMyBP-C$^{C1C7}$ removal, we used small-angle X-ray diffraction, a powerful method that leverages the partially crystalline arrangements of sarcomere proteins to evaluate their structural features under near-physiological conditions[10,36]. A focused, high-intensity X-ray beam from a synchrotron source passes through a fiber perpendicular to its long axis, producing a diffraction pattern on a detector. The intensities and spacings of the diffraction features (i.e., reflections; Fig. 2a) provide structural information regarding specific periodic features in the sarcomere, as described below. Generally, structural parameters can be variable between samples, which reduces statistical power to detect differences between conditions. However, our experimental approach of conducting all experimental treatments within one sample allows for the unusual ability to conduct repeated-measures statistical analyses that improved statistical power to assess the directionality of differences between conditions. The equatorial portion of the X-ray diffraction pattern provides information pertaining to the sarcomere lattice, while the meridional portion provides information regarding axial periodicities in the protein arrangement in the thick and thin filaments[10,37,38]. The SNOOPC2 X-ray experiments were conducted on SNOOPC2 psoas fiber bundles with detailed statistical assessments provided below and in Supplementary Tables 9 and 10. Control experiments with wildtype fiber bundles incubated with TEV$_P$ indicated no TEV$_P$ impact on structural features (Supplementary Fig. 2, Supplementary Table 11) or else discussed below. Finally, statistical analysis between control and SNOOPC2 datasets did not reveal any major structural differences, with further statistical details available in Supplementary Table 12.

We first considered a long-debated question in muscle physiology: are there load-bearing MyBP-C "C-links" between thick and thin filaments? A series of recent imaging studies in relaxed cardiac sarcomeres provide compelling evidence that MyBP-C can interact with thin filaments[29,30], but the strength of these interactions is unclear. The interdigitating thick and thin filaments of the sarcomere form a hexagonal lattice, where 6 thin filaments surround a thick filament (Fig. 2b). Mechanically speaking, in passive muscle, any protein linking the thick to thin filaments should pull on the thin filaments as SL increases, which would be expected to produce both radial forces that pull the filaments together and longitudinal forces that elongate the thin filaments. Therefore, we used our X-ray dataset to assess lattice spacing and thin filament length before and after fMyBP-C$^{C1C7}$ removal. The inter-filament lattice spacing in relaxed sarcomeres is a key determinant of force production because cross-bridge kinetics are sensitive to lattice spacing[39,40]. Lattice spacing is well known to be modulated by titin filaments, specifically the I-band spring that produces a compressive force on the myofilament lattice and centering the thick filaments within the sarcomere[16,41,42] but the role of C-links is unknown. Inter-filament lattice spacing can be derived from the 1.0 equatorial reflections, while the radial width of the peaks estimated via the radial width parameter, $\sigma_D$ (nm$^{-1}$), provides a measure of lattice spacing inhomogeneity[10] among myofibrils. Before treatment, increasing SL

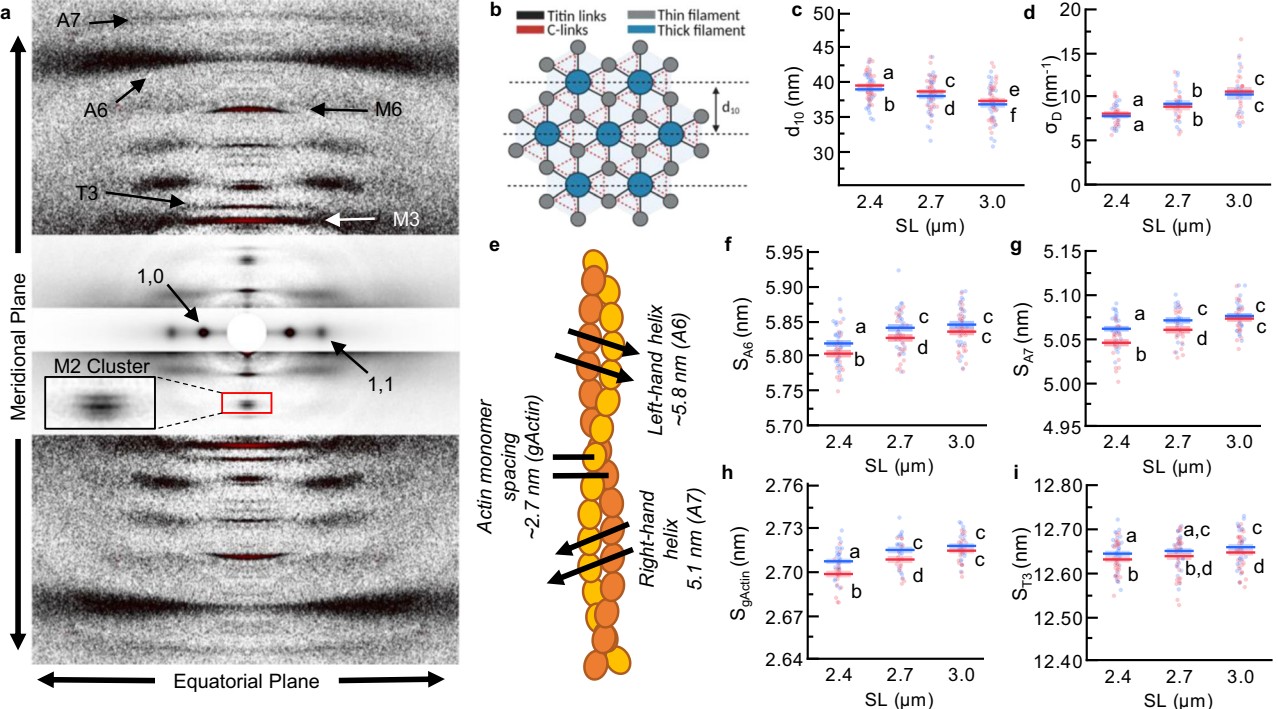

**Fig. 2 | Lattice and thin filament structural parameters of skeletal muscle fibers before (blue) and after (red) cleavage of the fMyBP-C bridge region across SLs.** **a** A representative X-ray diffraction pattern of permeabilized fiber bundles, with key reflections and axis orientation indicated. Three different intensity scales were overlayed so that features of interest could be best viewed by the eye. X-ray pattern after 1 s exposure, with sample at 2.4 μm SL before treatment. Exemplar 1D profile plots are provided in Supplementary Fig. 3. **b** Schematic of the sarcomere lattice, with titin, and MyBP-C C-link points indicated, as well as the $d_{10}$ lattice plane. C-links can attach to either of the two actin filaments nearby, and so this is depicted by the orange triangle. **c** Quantified $d_{10}$. **d** Quantified $\sigma_D$, a measure of the variability in the $d_{10}$ spacing. **e** Cartoon representation of actin, with important structural features

indicated. **f** A6 spacing ($S_{A6}$) from the right-handed actin helix. **g** A7 spacing ($S_{A7}$) from the left-handed helix of actin. **h** Actin monomer spacing ($S_{gActin}$), a measure of thin filament axial length. **i** T3 spacing ($S_{T3}$) from the troponin periodicity. Statistics throughout are ANOVA designs with main effects treatment, SL, and their interaction, and a random effect of individual, followed by Tukey's honestly significant difference (HSD) post-hoc test on statistically significant main effects ($P < 0.05$), and reported in figures as connecting letters: different letters are significantly different. Data throughout was reported as mean ± SE. Experimental dataset derived from 41 fiber bundles from psoas muscles of 15 SNOOPC2 mice (9 male/6 female). Further statistical details are in Supplementary Table 9.

decreased $d_{10}$ (Fig. 2c) and increased $\sigma_D$ (Fig. 2d) as previously reported[16,41]. In comparison, fMyBP-C$^{C1C7}$ removal did not alter $\sigma_D$ (Fig. 2d), but $d_{10}$ generally increased across SLs (Fig. 2c), suggesting that fMyBP-C$^{C1C7}$ indeed modulates the lattice spacing, in agreement with a previous report on fMyBP-C KO muscle[23].

We next turned our attention to SL-dependent thin filament elongation in relaxed sarcomeres, which has previously been observed in passively stretched cardiac and skeletal sarcomeres[16,43]. Here, we measured thin filament elongation (Fig. 3e) using the spacing of the left-handed helix ($S_{A6}$; at ~5.8 nm; Fig. 2f) and the right-handed helix ($S_{A7}$; at ~5.1 nm; Fig. 2g) in the actin filaments to estimate (see the "Methods" section) the axial spacing of actin monomers ($S_{gActin}$; ~2.7 nm; Fig. 2h). Before treatment, $S_{gActin}$ increased with SL as described previously[16], with thin filament extension observed from 2.4 to 2.7 μm SL with no further elongation from 2.7 to 3.0 μm SL (Fig. 2i). In comparison, fMyBP-C$^{C1C7}$ removal reduced $S_{gActin}$ at 2.4 and 2.7 μm SL, but not at 3.0 μm SL (Fig. 2h). However, these changes did alter the magnitude of thin filament elongation from 2.4 to 2.7 μm SL, while elongation also continued from 2.7 to 3.0 μm SL to match the 3.0 μm SL values before treatment. We additionally assessed the spacing of the third-order meridional reflection ($S_{T3}$; Fig. 2i) from the thin filament-bound troponin complex. $S_{T3}$ also increased with thin filament elongation following fMyBP-C$^{C1C7}$ removal, which could be completely accounted for by the stretching of the thin filament (ANCOVA covariate $P < 0.0001$, treatment $P = 0.28$; interaction $P = 0.93$), indicating no detectable structural changes in troponin apart from that imposed by

thin filament stretch itself. Therefore, as with lattice spacing, fMyBP-C$^{C1C7}$ does indeed play a role in dictating thin filament length.

From our findings, it would be logical to postulate that cleaving fMyBP-C$^{C1C7}$ terminates the thick-thin filament bridge and removes any MyBP-C-based pulling forces placed on it, which would both increase lattice spacing and shorten the thin filament. However, based on the statistically insignificant different passive tension drop after fMyBP-C$^{C1C7}$ removal (Fig. 1f), the magnitude of MyBP-C-based pulling forces on the lattice are most likely small and may not be able to account for these structural changes alone. Furthermore, while fMyBP-C$^{C1C7}$ removal decreased overall thin filament length, it did not eliminate thin filament elongation during passive sarcomere stretch. Therefore, it seems that parallel processes are occurring in the sarcomere. We posit two possibilities that are not necessarily mutually exclusive. First, the uncleavable slow-MyBP-C$^{C1C7}$, although few in psoas muscle, could nevertheless form C-links and contribute to stretch-dependent thin filament extension. If slow-MyBP-C$^{C1C7}$ contributes disproportionate forces on the lattice, then this could explain why passive tension did not change much after fMyBP-C$^{C1C7}$ removal and why thin-filament length still extended with sarcomere stretch. The second possibility relies on the known property that low-level, force-producing cross-bridges occur in passive muscle[44,45], and so can contribute to thin filament stretch. It is possible that fMyBP-C$^{C1C7}$ removal alters this behavior, as is supported by other studies, for example[23], and this study (also see structural data on myosin heads detailed below), which show both

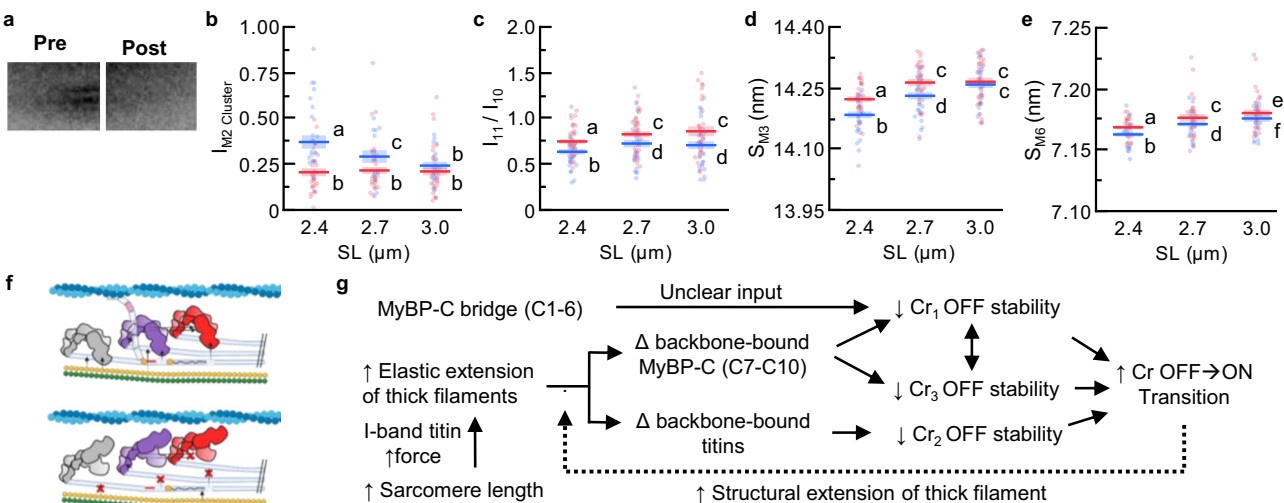

**Fig. 3 | Myosin ON/OFF structural parameters of skeletal muscle fibers before (blue) and after (red) cleavage of the fMyBP-C bridge region across SLs. a** M2 cluster before and after fMyBP-C cleavage. **b** Intensity of the M2 cluster ($I_{M2}$) cluster. **c** $I_{1,1}/I_{1,0}$ provides a measure of the mass distribution between the thick and thin filaments. **d** M3 spacing ($S_{M3}$) measures the axial spacing of myosin heads. **e** M6 spacing ($S_{M6}$) is the spacing of the thick filament backbone and is often used as a measure of thick filament length. **f** Representation of myosin OFF–ON transition before (top) after (bottom) N-terminal fMyBP-C cleavage. We predict that a loss of regulation of N-terminal MyBP-C leads to an OFF→ON transition on $Cr_1$ (purple) myosins and has a cooperative effect on surrounding $Cr_2$ and $Cr_3$ myosins to also transition OFF → ON. **g** Based on mechanical and structural details from the

current and other studies, we present a flow chart of myosin head ON/OFF control. This chart shows how myosin heads transition OFF → ON, but we assume the reverse effects will also transition ON → OFF. Statistics throughout are ANOVA designs with main effects treatment, SL, and their interaction, and a random effect of individual, followed by Tukey's honestly significant difference (HSD) post-hoc test on statistically significant main effects ($P < 0.05$), and reported in figures as connecting letters: different letters are significantly different. Data throughout was reported as mean ± SE. Experimental dataset derived from 41 fiber bundles from psoas muscles of 15 SNOOPC2 mice (9 male/6 female). Further statistical details are in Supplementary Table 10.

altered cross-bridge kinetics and loss of myosin head suppression when MyBP-C is altered. The potential interplay between these scenarios, most likely running in parallel and somewhat interconnected, makes it difficult to study each individually but could potentially be separated out using computational modeling and is worth exploring[46].

## fMyBP-C$^{C1C7}$ cleavage alters the conformation of myosin heads

Of note for this study is a grouping of up to four X-ray meridional reflections related to the presence of MyBP-C in the so-called M2 cluster[38,47] that are not present in transformed electron micrographs, analogous to X-ray diffraction patterns from MyBP-C KOs[48]. In the present study, we did not typically have sufficient resolution to separate out the four reflections, so instead, we measured the intensity of this entire M2 cluster[49] ($I_{M2 cluster}$; Fig. 3a) before and after removal of fMyBP-C$^{C1C7}$. We report that prior to treatment, increasing SL decreased the $I_{M2 cluster}$. In contrast, fMyBP-C$^{C1C7}$ removal substantially decreased the $I_{M2 cluster}$ at shorter SLs to a value that was independent of SL (Fig. 3a, b). While the M2 cluster is clearly associated with the presence of fMyBP-C$^{C1C7}$, the reflections are most likely not due to MyBP-C molecules themselves because MyBP-Cs are relatively sparse compared to other repeating proteins (i.e., myosins), and so would be expected to create a reflection too weak to be measured[37]. Instead, the M2 cluster reflections most likely arise from a sub-population of myosin heads that interact with MyBP-C in the C-zone, which causes them to be perturbed and generate their own periodicities (e.g. the M2 cluster)[37,50]. In complementary evidence, high-resolution cryo-EM and cryo-ET structures of cardiac thick filaments in the OFF state indicate interaction and potential cooperativity of myosin heads with other myosin heads and MyBP-C[29,31,51], providing a pathway for relatively few MyBP-Cs to impact many myosin head ON/OFF states. In summary, our data supports the notion that the fMyBP-C$^{C1C7}$ contributes to the M2 cluster reflections by binding and perturbing a subpopulation of myosin heads.

There are structural signatures that provide evidence for MyBP-C stabilizing the OFF conformation of the myosin heads in the C-zone[4,23,24] as previously proposed[29]. The equatorial intensity ratio ($I_{1,1}/I_{1,0}$) is often used to quantify the myosin head OFF/ON state transitions. It provides a measure of the transfer of mass (i.e., myosin heads) from the thick to the thin filaments, with increasing $I_{1,1}/I_{1,0}$ indicating that more myosin heads are in a favorable orientation to associate with the thin filaments upon contraction[52]. In additional complementary evidence, myosin head configuration can be evaluated via the spacing of the M3 myosin meridional reflection ($S_{M3}$), which reflects the average distance of ~14.3 nm spacing between the myosin crowns along the thick filament. Increases in $S_{M3}$ are typically associated with a subpopulation of myosin heads moving from the OFF to the ON state or the reorientation of myosin heads to a more perpendicular orientation relative to the thick filament backbone that increases the chance of attachment[10,53]. We report that, before fMyBP-C$^{C1C7}$ removal, $I_{1,1}/I_{1,0}$ (Fig. 3c) and $S_{M3}$ (Fig. 3d) shifted to larger values when fibers were stretched from 2.4 to 2.7 μm SL, indicating that a proportion of myosin heads shifted from the OFF to the ON state in response to passive stretch in agreement with previous work[16]. After N-terminal fMyBP-C cleavage, both $I_{1,1}/I_{1,0}$ and $S_{M3}$ were elevated at 2.4 and 2.7 μm SL but still followed the same length-dependent trend as before cleavage. Since the structural and mechanical signatures (see above) of LDA are still present after fMyBP-C cleavage, we conclude that other mechanisms dominate in LDA, such as those involving the fMyBP-C C-terminal domains (C8–C10)[29] and strain generated in the thick filaments by titin-based passive tension[16,41,43].

Compared to the length-dependent increase of $I_{1,1}/I_{1,0}$ and $S_{M3}$ from 2.4 to 2.7 μm SL, there is no detectable increase from 2.7 to 3.0 μm SL (Fig. 3c, d). However, this may not necessarily suggest that the LDA effect has reached a maximum. The decrease in thick-thin filament overlap associated with increasing the SL from 2.7 to 3.0 μm SL can itself decrease $I_{1,1}/I_{1,0}$, regardless of myosin head movement[54], while myosin heads in the ON state potentially become disordered and may

contribute less to $S_{M3}$ compared to OFF myosin, pushing values down. Therefore, the probable structural scenario represented by the data is that more myosin heads are moving radially towards the thin filament with increasing SL between 2.4 and 2.7, as well as from 2.7 and 3.0 μm SL (OFF-to-ON transitions). Taken together, our data support the notion that loss of fMyBP-C$^{C1C7}$ reduces stabilization of the OFF state of myosin heads, resulting in the release of some myosin heads from the OFF to the ON state. This would also explain the results of MyBP-C phosphorylation, which also leads to OFF-to-ON transitions of myosin heads[24,55] and may be caused by a phosphorylation-dependent destabilization of MyBP-C and myosin heads.

### fMyBP-C$^{C1C7}$ cleavage increases thick filament length

We next focused on the elongation of thick filaments associated with sarcomere-stretch in passive muscle, a property correlated with increasing OFF-to-ON myosin head transitions that have been proposed to contribute to LDA in some systems[10,12,16]. Elongation of the thick filaments during relaxed sarcomere stretch is predominately caused by titin-based forces, which pull on the tips of the thick filaments and increase with increasing SL[16,41]. Thick filament extension can be quantified by the spacing of the M6 reflection ($S_{M6}$), which tracks the ~7.2 nm periodicity that arises off the thick filament backbone, with increasing $S_{M6}$ indicative of thick filament extension[56,57]. Before and after fMyBP-C$^{C1C7}$ loss, we observed the expected increase in $S_{M6}$ with stretch from short to long SLs (Fig. 3e). Following cleavage of fMyBP-C; however, $S_{M6}$ was increased at every SL, suggesting that fMyBP-C$^{C1C7}$ affects thick filament structure independent of the titin-based forces placed on it. Of note, the regression slope of $S_{M6}$ versus SL is similar between treatment conditions (treatment*SL $P = 0.62$), suggesting that changes in filament stiffness do not play a role in thick filament structural changes in this case.

This begs the question: what drives this filament stiffness- and SL-independent change in thick filament length? Recent studies provide evidence that simply shifting a subset of myosin heads between OFF and ON states via chemical treatment of relaxed fibers can change thick filament length[12,38,58,59]. For example, incubation with the drug mavacamten transitions myosin heads from the ON to the OFF state and is accompanied by a shorter $S_{M6}$[12], while incubation with 2′-deoxy-ATP (dATP) transitions myosin heads from the OFF to the -ON state and is accompanied by a longer $S_{M6}$[58]. Our data indicate that loss of fMyBP-C$^{C1C7}$ promotes OFF-to-ON transitions (increases in $I_{11}/I_{10}$ and $S_{M3}$), and so this would also track with changes to thick filament length. The idea that MyBP-C can regulate the OFF-to-ON transition of myosin heads has been discussed for some time[10,13,60], but our data (Fig. 3c–f) combined with recent sarcomere and thick filament reconstructions[29,31] provide sufficient detail to suggest a possible mechanism (Fig. 3g). As detailed in Fig. 3g, our data are consistent with the hypothesis that the loss of fMyBP-C$^{C1C7}$ promotes OFF-to-ON transitions of $Cr_1$ myosin heads in the C-zone. This is proposed to create localized elongations of the thick filament backbone (which we observe) that affect nearby OFF $Cr_2$ and $Cr_3$ myosin heads, which then break their docked formations and transition from OFF-to-ON states. One critical detail to uncover is why this purported feed-forward mechanism does not continue until all myosin heads are in the ON state, implying the existence of some type of as-of-yet unknown molecular brake to limit OFF-to-ON state changes.

### Unifying scheme for thick filament ON-to-OFF transition

Combining our data with OFF-to-ON transition data in titin[16], myofilament protein location, and interaction data from recent high-resolution cryo-ET experiments[29,31,51], and decades of accumulated data and hypotheses regarding the functional roles of MyBP-C[4], we propose a unifying scheme for sarcomere OFF-to-ON transitions (Fig. 3f, g). Sarcomere stretch produces titin-based forces that elongate the thick filament backbone, leading to strain in the thick filament-

bound titin and MyBP-C domains. This disrupts the myosin OFF-state, namely those involving $Cr_2$ (i.e. titin interactions[29] across the whole filament) and $Cr_1$ and $Cr_3$ interactions in the C-zone (i.e. MyBP-C C8 and C10, respectively[29,31]). Separately, N-terminal or middle MyBP-C domains act to maintain myosin heads in the OFF state, but when perturbed (in our case, removed), there are OFF-to-ON transitions of $Cr_1$ myosin heads, which are thought to then indirectly destabilize $Cr_3$ myosin OFF states via the stabilizing interactions of $Cr_1$ or $Cr_3$ myosins[51]. Furthermore, OFF-to-ON transitions of $Cr_1$ lead to a sarcomere-length independent structural elongation of the thick filament[12,13,58], which perturbs thick-filament bound titin and MyBP-C domains, potentially destabilizing their interaction with docked myosin heads. It should also be noted that myosin, MyBP-C, and titin can all be affected in ways that impact their role on OFF−ON myosin states by mutations or by local environmental factors such as phosphorylation, oxidation and pH−all of which can occur in disease but also are common in healthy people during exercise[61–63]. Therefore, there may be many ways to fine-tune OFF-to-ON state transitions in skeletal muscle to meet performance demands in real-time and provide plasticity in health and disease.

### Limitations and outlook

Known limitations of the TEV protease procedure are that membrane permeabilization can expand the sarcomere lattice while experiments at 25 °C naturally reduce myosin head order and partially shift heads toward the ON state. Nevertheless, there is no a priori reason to expect that the directionality of the effects of MyBP-C cleavage on the myofilaments would be different from in vivo conditions. Therefore, identifying a strategy to rapidly cleave fMyBP-C$^{C1C7}$ using intact preparations is an immediate next goal, which would then also provide more detailed reflection information, such as myosin layer lines 1 and 4, and higher order reflections that would further illuminate the orientation of myosin heads within the sarcomeres. We are therefore excited that technology is now poised to explore MyBP-C function in otherwise healthy, living sarcomeres and is a promising new approach within muscle physiology that will only improve going forward.

## Methods

### Animal model and muscle preparation for X-ray experiments

Animal procedures were approved and performed according to the guidelines of the local animal care and use committee (IACUC) of the University of Arizona. Both male and female mice were used in this study design, as we have previously found no sex-based differences in the TEV protease treatment effect in skeletal fibers[16]. SNOOPC2 mice were bred and housed at the University of Arizona. Genotyping was performed via PCR analysis and protein gels used to assess TEV protease reactivity evaluated by measuring myosin binding protein C (MyBP-C) cleavage before and after treatment. Genetically homozygous and wildtype adult SNOOPC2 mice (age range, 2–6 months) were humanely euthanized; psoas muscle was immediately extracted for long-term storage and permeabilized ("skinned") for use in X-ray experiments at −20 °C using standard glycerol techniques (1:1 rigor: glycerol; rigor contains (in mM) KCl (100), MgCl$_2$ (2), ethyleneglycol-bis(β-aminoethyl)-N,N,N′,N′-tetraacetic acid (EGTA,5), Tris (10), dithiothreitol (DTT, 1), protease inhibitors [Complete, Roche Diagnostics, Mannheim, Germany], pH 7.0). Samples were shipped to the BioCAT facility on ice for all experimental tests and stored at −20 °C until used. On the day of experiments, psoas muscles were removed from the storage solution and vigorously washed in relaxing solution (composition (in mM): potassium propionate (45.3), N,N-Bis(2-hydroxyethyl)-2-aminoethanesulfonic acid BES (40); EGTA (10), MgCl$_2$ (6.3), Na-ATP (6.1), DTT (10), protease inhibitors [Complete], pH 7.0)). Bundles containing 10–20 fibers (3–6 mm long, fiber bundle diameter: 0.49 ± 0.02 mm) were carefully excised and kept in the physiological

register by tying silk suture knots (sizing 6–0 or 4–0) at the distal and proximal ends of the bundle. Samples were then immediately transferred to the experimental chamber (see below).

## N-terminal fMyBP-C cleavage protocol

All experiments were conducted by running the below mechanical experiments before and after incubation with tobacco etch virus (TEV) protease, which selectively cleaves the TEV protease recognition site of fast-isoform MyBP-C in SNOOPC2 muscle so that the N-terminal region detaches and diffuses out of the sarcomere (Fig. 1; Supplementary Fig. 1), similar to that designed previously for cardiac MyBP-C[64]. The samples were incubated with $TEV_P$ for 30 min. Recombinant TEVp was purified[64] or was purchased from Thermo Fisher Scientific, USA, and used at 100 units $acTEV_P$ in 300 µl relaxing solution. After incubation, fibers were rinsed in a fresh relaxing solution to remove excess protease.

## Western blot

Proteins were prepared for western blotting by pulverizing psoas muscle tissue from both WT and HOM SnoopC2 mice with a pestle and mortar cooled with liquid nitrogen. Samples were then homogenized using a Polytron Homogenizer (PT1200E, Kinematica, Switzerland) in a skinning solution ([in mmol/L]: 5.92 $Na_2ATP$, 6.04 $MgCl_2$, 2 EGTA, 139.6 KCl, 10 imidazole with 0.01% saponin, 1% Triton X-100®, and Halt protease inhibitor cocktail, EDTA-free [78437, ThermoFisher Scientific, USA], pH 7.0). The homogenized tissue was then tumbled for 15 minutes at 4 °C and washed in a relaxing solution ([in mmol/L]: 5.92 $Na_2ATP$, 6.04 $MgCl_2$, 2 EGTA, 139.6 KCl, 10 imidazole). Tissue was then either left untreated, treated with TEV protease (12 µg protease per mg tissue) for 30 min at room temperature, or treated with TEV protease for 30 min at room temperature, and then washed in relaxing solution to remove cleaved protein. The tissue in relaxing solution was then mixed with an equal volume of urea buffer ([in mol/L]: 8 urea, 2 thiourea, 0.05 Tris–HCl, 0.075 dithiothreitol with 3% SDS and 0.03% bromophenol blue, pH 6.8), run on an SDS–PAGE gel (4561086, 4–15% Mini-PROTEAN® TGX™ Precast Protein Gel, Bio-Rad), and transferred onto a nitrocellulose membrane. Blots were blocked with OneBlock™ Fluorescent Blocking Buffer (20-314, Genessee Scientific) and stained for either fMyBP-C (MYBPC2 polyclonal rabbit antibody diluted 1:2000, PA5-83638, ThermoFisher Scientific, USA) or sMyBP-C (MYBPC1 polyclonal rabbit antibody diluted 1:1000, NBP2-41157, Novus Biologicals) with actin (actin monoclonal mouse antibody diluted 1:2000, ACTN05 [C4] MA5-11869, ThermoFisher Scientific, USA) as a loading control. Secondary antibodies used were goat anti-rabbit IRDye 800CW (926-32211, LI-COR) and goat anti-mouse IRDye 680RD (926-68070, LI-COR).

## Crossbridge kinetics/mechanics

For mechanical measurements in permeabilized psoas from SNOOPC2 mice, muscle fibers were isolated by gentle mechanical disruption (2–3 s at a low setting using a Polytron Homogenizer PT1200E, Kinematica, Switzerland) in a skinning solution (5.92 mM $Na_2ATP$, 6.04 mM $MgCl_2$, 2 mM EGTA, 139.6 mM KCl, 10 mM imidazole, 0.01% saponin, 1% Triton-X-100®, Halt protease inhibitor cocktail, EDTA-free [78437, Thermo Fisher Scientific, USA]). Following the mechanical disruption, fibers tumbled for 30 minutes at 4 °C to allow for the removal of the sarcoplasmic reticulum, sarcolemma, and any remaining endogenous $Ca^{2+}$, leaving intact the myofibrillar network. Psoas fibers were then thoroughly washed in solution without detergents. Psoas fibers (~100–250 µm in length) were attached between a high-speed motor (Model: 315C-I, Aurora Scientific Inc., Aurora, Ontario, Canada) and a force transducer (Model 403 A series, Aurora Scientific Inc., Aurora, Ontario, Canada) with an aquarium sealant (Marineland, 100% clear silicone rubber). The motor and force transducer were mounted above a temperature-controlled platform (Model 803B, Aurora Scientific Inc., Aurora, Ontario, Canada) regulated by a thermocouple (825A, Aurora

Scientific Inc., Aurora, ON, Canada) located on the stage of an inverted microscope (Model IX-53, Olympus Instrument Co., Japan). The glue was allowed to cure for 30 minutes before the start of the experiment. Using push-button micromanipulators, sarcomere length (SL) was manually set to either 2.4, 2.7, or 3.0 µm under passive conditions. SL was determined from video analysis, and force measurements were made by activating myocytes in pCa solutions containing variable free calcium concentrations ranging from pCa 9.0 to 4.5. All measurements were taken at 15 °C. Isometric force measurements ($F$) were normalized to maximal force ($F_0$) at pCa 4.5 and to the cross-sectional area of the muscle preparation, assuming circular dimensions. Data was plotted in GraphPad Prism and fitted using a sigmoidal four-parameter logistic curve. The $pCa_{50}$ is the concentration of $Ca^{2+}$ required to achieve half-maximal activation of the myocyte. The rate of force redevelopment ($k_{tr}$) was calculated by fitting force traces with a single exponential curve and normalizing the rates (k) to the maximal rate of force redevelopment ($k_O$) measured at pCa 4.5. All rates were then plotted against isometric force. Passive force values were measured in a range of SL (2.2, 2.4, 2.6, 2.8, and 3.0 µm) and plotted as absolute values or normalized to the maximal passive force measured at SL 3.0 µm within the treatment condition.

## Small-angle X-ray diffraction experiments

X-ray diffraction patterns were collected using the small-angle instrument on the BioCAT beamline 18ID at the Advanced Photon Source, Argonne National Laboratory. The X-ray beam (0.103 nm wavelength) was focused to -0.06 × 0.15 mm at the detector plane, with an incident flux of $-3 \times 10^{12}$ photons per second. The sample-to-detector distance was set at -2 m, and the X-ray fiber diffraction patterns were collected with a downstream CCD-based X-ray detector (Mar 165, Rayonix Inc., USA). Muscle preps were hung on custom muscle mechanics rigs between a force transducer (402A, Aurora Scientific, Canada) and length motor (322C, Aurora Scientific, Canada), with force and length data collected at 1,000 Hz using a 600A: Real-Time Muscle Data Acquisition and Analysis System (Aurora Scientific, Canada)[16]. Diffraction patterns were captured with 1 s exposure times. An inline camera allowed for initial alignment with the X-ray beam and continuous visual quality control of samples during the experiment (e.g., check for fiber slippage and preparation damage). SL was measured via laser diffraction using a 4-mW Helium–Neon laser at 2.4 µm SL, with an SL variability of about ±0.05 µm SL, based on the width of the first-order diffraction. The force baseline was set at slack length. After this initial setup, fiber length changes were accomplished through computer control of the motor, where studies before beamline experiments confirmed appropriate SL change. Experiments were conducted at room temperature (-25 °C). The mechanical rig was supported on a custom-designed motorized platform that allowed placement of muscle into the X-ray flight path and small movements to target X-ray exposure during experiments. Using the inline camera, the platform was moved to target the beam at different locations along the length of the sample. To limit X-ray exposure of any one part of the preparation, no part of the sample was exposed more than once. Furthermore, we previously conducted controls on mouse psoas where experiments were run as described here but without $TEV_P$ incubation, and experimental values from the "pre" to "post" trials were consistent[16]. Here, we further assessed the effect of $TEV_P$ exposure on WT psoas fiber bundle preparations (Supplementary Fig. 2; Table 7) and found no difference between pre- and post-treatment experimental values. Fiber bundle diameter was measured using the inline camera, and physiological cross-sectional area was calculated at the initial fiber length, with the assumption that the sample was a uniform cylinder longitudinally.

The mechanical protocol during these experiments consisted of passive ramp-hold stretches. Relaxed samples started at 2.4 µm SL, at which an X-ray diffraction pattern was collected, then stretched (over 60 s) to 2.7 µm SL, held for 60 s before another X-ray diffraction

pattern collection, and then similarly stretched to 3.0 μm SL and held for a final X-ray diffraction pattern collection. Samples then underwent the TEV protease protocol at 2.4 μm SL as explained above, and the mechanical experiment was repeated.

## X-ray diffraction pattern analysis

X-ray diffraction patterns were initially reduced and prepared for analysis using Bulb (Accelerated Muscle Biotechnologies, Boston, USA) and then analyzed using the MuscleX open-source data reduction and analysis package (BioCAT, Argonne National Laboratory). The Quadrant Folding routine was used to improve the signal-to-noise by adding together the four equatorial-meridional quadrants, which each provide the same information (Friedel's Law), followed by the built-in circularly symmetric background subtraction protocol. Reflection intensity values were divided by the total subtracted background to normalize all images. The Scanning Diffraction routine was used to measure the angular divergence of the 1,0 equatorial reflection. The routine obtains 2D and 1D radially integrated intensities of the equatorial intensities and then fits Gaussian functions over the diffraction peaks to calculate the standard deviation (width $\sigma$) intensity distribution pattern. In this process, the routine obtains the integrated intensity of each equatorial reflection as a function of the integration angle. The Equator routine of MuscleX was used to calculate the $I_{1,1}/I_{1,0}$ intensity ratio, $d_{10}$ lattice spacing between thick filaments, and $\sigma_{D}$, a measure of the variability in thick filament lattice spacing (a proxy for lattice ordering). The equatorial intensities (1,0 and 1,1) of the X-ray pattern were determined by integrating the intensities along the equatorial axis ($\pm 0.005$ nm$^{-1}$ on either side of the equatorial axis), followed by 1,0 and 1,1 fitting Gaussian curves to the 1,0 and 1,1 reflection, respectively. Spacing and intensities were calculated from the maximum Gaussian peak and area under the Gaussian curve, respectively. There are limitations to the $I_{1,1}/I_{1,0}$ in protocols that disrupt lattice order, such as cleaving titins[16], because it is not easy to uncouple the effects of lattice disorder[65] and mass shift due to myosin head movement on $I_{1,1}/I_{1,0}$. In the current model, cleaving N-terminal MyBP-C does not show effects on lattice order (e.g. $\sigma_{D}$), and so we considered $I_{1,1}/I_{1,0}$ a trustworthy assessment of the average movement of myosin head mass. Meridional ($I_{M2\_Cluster}$, $S_{M3}$, $S_{T3}$, $S_{M6}$) and off-meridional reflections ($S_{A6}$, $S_{A7}$) were collected using the MuscleX routines diffraction centroids and projection traces. The meridian intensities along the meridional axis ($I_{M2\_Cluster}$, $S_{M3}$, $S_{T3}$, $S_{M6}$) were first calculated by integrating in the reciprocal radial range ~$0 \leq R \leq 0.032$ nm$^{-1}$ for M3, M6, and T3 reflections, and ~$0.013 \leq R \leq 0.053$ nm$^{-1}$ for the A6 and A7 reflection, where $R$ denotes the radial coordinate in reciprocal space[66]. Next, integration limits along the median axis were as follows: M2 cluster $0.04 \leq R \leq 0.05$ nm$^{-1}$; M3, $0.066 \leq R \leq 0.074$ nm$^{-1}$; M6, $0.136 \leq R \leq 0.144$ nm$^{-1}$; T3, $0.078 \leq R \leq 0.080$ nm$^{-1}$; A6, $0.166 \leq R \leq 0.175$ nm$^{-1}$; A7, $0.192 \leq R \leq 0.204$ nm$^{-1}$. From these, the spacing and intensity of each reflection are calculated automatically. The $S_{A6}$ and $S_{A7}$ report on the left- and right-handed actin helical structures within the thin filament were used here to calculate the axial spacing of the actin monomers ($S_{gActin}$; Eq. (1)), where $S_{gActin}$ can be used as a measure of thin filament extension[66,67].

$$gActin\ spacing = \frac{1}{\left(\frac{1}{A6\ Spacing}\right) + \left(\frac{1}{A7\ Spacing}\right)} \quad (1)$$

Every image provides reflections of different quality, which leads to various levels of Gaussian fit errors for each reflection modeled, increasing variation in spacings in the dataset. To limit these effects, fit errors > 10% were discarded. Positions of X-ray reflections on the diffraction patterns in pixels were converted to sample periodicities in nm using the 100-diffraction ring of silver behenate at $d_{001} = 5.8380$ nm.

## Statistics

Statistical analysis was conducted using JMP Pro (V16, SAS Institute Inc., Cary, NC, USA). The significance level was $\alpha = 0.05$. Response variables included all X-ray and mechanics parameters. We first built a repeated-measures analysis of variance (ANOVA) design. We used fixed effects treatment (pre-/post-TEV protease incubation) and SL, a treatment × sarcomere interaction term, and a random (repeated-measures) effect of individuals. Data were best Box-Cox transformed to meet assumptions of normality and homoscedasticity, when necessary, which were assessed by residual analysis, Shapiro–Wilk's test for normality, and Levene's test for unequal variance. Statistically significant main effects were subject to Tukey's highly significant difference (HSD) multiple comparison procedures to assess differences between factor levels. These data were indicated in graphs via so-called connecting letters, where factor levels sharing a common letter are not significantly different from each other. Data is presented at mean ± standard error of the mean (SE) unless otherwise noted.

## Reporting summary

Further information on research design is available in the Nature Portfolio Reporting Summary linked to this article.

## Data availability

All data are available in the main text or the supplementary materials or available upon reasonable request. Source data are provided with this paper.

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

## Acknowledgements

This research used resources of the Advanced Photon Source (APS), a U.S. Department of Energy (DOE) Office of Science User Facility operated for the DOE Office of Science by Argonne National Laboratory under Contract No. DE - AC02 - 06CH11357, and further NIH support. The content is solely the responsibility of the authors and does not necessarily reflect the official views of the National Institute of General Medical Sciences or the National Institutes of Health. We thank the BioCAT beamline support staff at the APS. We thank Dr. Massimo Reconditi for insightful technical discussion and Anna Good for artistic design and editing. BioRender.com was used to construct some of the figure cartoons. Funding for this study was provided by the German Research Foundation (454867250 [ALH], SFB1002A08 [WAL]), IZKF Münster (Li1/029/20 [WAL]), National Institute of Health (P41 GM103622 [TCI]), P30 GM138395 [TCI], HL080367 [SPH]), HL140925 [SPH], AR081935 [SPH], T32 HL007249 [NME], American Heart Association (827628 [NME]).

## Author contributions

A.L.H. and S.P.H. conceptualized the project; A.L.H., W.M., N.M.E., R.L.S., S.P.H. developed the methods; A.L.H., N.M.E., R.L.S., D.N., M.K., S.P.H. conducted the investigation; A.L.H., N.M.E., M.K. visualized the datasets; A.L.H., S.P.H., T.C.I., W.A.L.; A.L.H. was the project administrator; A.L.H., S.P.H. supervised the study; A.L.H. wrote the original draft; all authors reviewed, edited, and agreed to the final draft.

## Funding

## Competing interests

T.I. provides consulting and collaborative research studies to Edgewise Therapeutics Inc. and A.L.H. and M.K. are owners of Accelerated Muscle Biotechnologies Consultants LLC. All other authors declare no competing interests.
