## [Peer Review File · Nature Communications]

Myosin-binding protein C regulates the sarcomere lattice and stabilizes the OFF states of myosin headsREVIEWER COMMENTS

Reviewer #1 (Remarks to the Author):

The manuscript describes experiments aimed at understanding the role of myosin binding protein-C (MyBP-C) in the contraction of skeletal muscle, and in particular to determine the interactions of MyBP-C with both thin and thick filaments within the myofilament lattice in-situ. The authors developed a SNOOPC2 homozygous mouse line to induce specific cleavage of the N-terminal region of MyBP-C, the C1C7 domains, via TEV protease in demembranated muscle fiber bundles from fast-twitch psoas skeletal muscle. From mechanical experiments on these samples, the authors conclude that MyBP-C cleavage decreases calcium-sensitivity of force and modulates crossbridge cycling kinetics at short sarcomere length. TEV cleavage did not affect passive force. From X-ray diffraction experiments performed in relaxed psoas bundles at different sarcomere lengths, the authors show that cleavage of MyBP-C C1C7 causes expansion of the myofilament lattice and extension of the thin and thick filaments, accompanied by disordering of the myosin heads, suggesting that MyBP-C forms C-links with the actin filament and is important to stabilize myosin OFF state. The idea behind the experiments is logical and clear and the experimental approach should allow to have paired controls before and after TEV cleavage of MyBP-C C1C7 removing some of the variability between samples. However, I have few questions on the methodology and data interpretation.

Major points:

- In the main text (second paragraph of page 3) the authors state that mechanical data were recorded “in permeabilized fiber bundles before and after cleavage...”; however, in methods, beginning of page 14, they say that “For mechanical measurements in permeabilized psoas from SNOOPC2 mice, psoas myocytes were mechanically homogenized... Psoas cells were attached...” Could the authors clarify what samples are used for the mechanical experiments? Muscle fibers cannot be prepared by homogenization.
- The authors show that psoas muscle fiber bundles (myocytes?) have a big LDA response, especially at pCa 4.5: the force measured at sarcomere length (SL) $2.7\mu\text{m}$ is more than double that developed at $2.4\mu\text{m}$ SL. This is not consistent with the tension-SL relationship of skeletal muscle.
- During the X-ray experiments, SL was measured by laser diffraction in bundles of 10-20 fibres. Following from the previous point, what is the error in the SL measurement using laser diffraction? Did the authors measure SL before X-ray exposure at each sarcomere length? The high spread in d10 (35-42 nm at $2.4\mu\text{m}$ SL before TEV in Fig.2C) is indicative of a big error in the SL measurement. This might also affect the observed changes in the other X-ray reflections between 2.4 - $2.7\mu\text{m}$ SL.
- N numbers for mechanical and X-ray data (number of mice and number of fiber bundles/myocytes(?)) are not specified in methods or in the figure legends.

Minor points:

- The authors say that the data in Fig 1L-M have been normalized by pCa 4.5, however data at pCa 4.5 are not shown in Fig 1G-I.
- The methods state that k_{tr} is measured as mono-exponential fit of force redevelopment. Could the authors explain what the dashed lines in Fig 1J-K represent? Please add ticks to the y-axis of panels J-K. The changes in % length change applied during the release-restretch protocol should be shown.
- The fiber bundles were exposed 3 x 1s at three increasing SL, then the protocol was repeated after TEV cleavage (30 min). To estimate the effect of TEV/radiation damage on wildtypes the authors only use the shorter SLs (2.4-2.7 μ m). How can the authors exclude that stretching to 3 μ m might trigger an irreversible change per se?
- Is the X-ray diffraction pattern in Fig. 2A an example from a single 1s exposure, or is it from added psoas bundles? Add exposure time and SL. It is impressive that it was possible to analyze data from each psoas fiber bundle independently, implying that data have good S:N. Could the authors add a supplementary figure showing an example of the 1D profiles (equatorial and meridional) from a single bundle at the three SLs before and after TEV cleavage.
- Page 2, 5 lines from bottom of page. "Free-heads" was not defined earlier. Is this really needed here?
- Page 3 first line. Could you clarify the calculation used to obtain the number of myosin heads "under the control of MyBP-C"?
- Fig 2A. M2 cluster magnification: the general reader would benefit by identifying with a rectangle the region of the 2D pattern that was magnified.
- Legend of Fig.2F-G. Left- and right-handed actin helix are inverted with respect to the cartoon in Fig 2E and the main text.
- Fig 2G. Check ticks of y-axis.
- Page 4 first paragraph. Could the authors discuss the result that C-links might produce radial and longitudinal forces, but have no effect on passive force (Fig 1F)?
- Fig. 3A should show changes in the M2 cluster within the 2D pattern: could the 2D magnification have the same orientation as in Fig.2A? Is the equator at the top in Fig 2A and at the bottom in Fig. 3A? In Fig 2A the magnification at the level of the M2 cluster includes the second order of the troponin reflection. Was the troponin reflection included within the integration limits used to calculate the intensity of the M2 cluster? Could the authors provide the integration limits they have used (not reported in methods)?
- Page 5, second paragraph. "..., with increasing I11/I10 indicating that more myosin heads are associated with the thin filaments". This sentence should be rephrased as the experiments were performed in relaxing conditions and the changes in I11/I10 are small compared to the changes observed between resting-active muscle (see your cited literature).
- Page 5, second paragraph, line 5-6: "myosin head configuration can be evaluated via the spacing of the M3 myosin meridional reflection (SM3)": this is an extrapolation. SM3, as you say later, is a measurement of the axial spacing of the myosin heads. To measure changes in heads conformation the authors might need to also measure changes in the intensity of the M3 reflection and in its fine structure, and a model.
- Page 5, second paragraph, line 10, and 2 lines from bottom: "behaved as expected" please rephrase, e.g. in agreement with previously published work.
- Model of thick filament activation, page 6 and Fig.3 G. The proposed model is plausible but will need further validation. As the authors include in their model the information from the recent cryo-EM structures (Dutta et al., Tamborrini et al.), could they discuss if and how the presence of the TEV

protease cleavage site and SnoopTag site between C7-C8 of MyBP-C might perturb the interactions with Crown 1 or Crown 2 tails per se?

- Related to the previous point, could the authors run a non-paired t-test to compare the data from X-ray signals obtained before TEV in control wildtype and homozygous SNOOPC2 psoas bundles (e.g. data in Extended data Fig.2 with data in Fig. 2 and 3)?
 - Page 12. Legend of Fig.3D: rephrase. M3 spacing SM3 measures the axial spacing of the myosin heads. Fig 3E legend: M6 spacing SM6 measures the spacing of the thick filament backbone.
 - Page 13. Western blot, 2nd line. “left ventricle tissue was homogenized..” why the authors used left ventricle tissue when the rest of the experiments were performed on psoas muscle?
 - Page 14. Small-angle X-ray diffraction experiments. 6th line: “For TC experiments...” please expand TC. Bottom line: could you report measured diameter of fiber bundles?
 - Page 15. X-ray image analysis. 9 lines from bottom. “Meridional (IM2cluster, SM3, IM3,..” IM3 is not shown. Could the authors add integration limits for the X-ray reflections analyzed (equatorial and meridional)?
 - Could the authors also report integration limits for the actin layer lines and the formula used to calculate Sgactin?
 - Extended data tables: Could you explain all used parameters in the legend? (e.g. does Tn-pCa stands for tension-pCa ?).
 - Ext data table 2: What is the absolute value of ktr?
- Ext data table 3-4: What are the units of measurement of absolute force?
- Ext data table 6: Are the mean numbers associated to IM2cluster normalized values? If yes, what would correspond to IM2cluster=1?

Reviewer #2 (Remarks to the Author):

This is an exciting time in muscle research with the recent BioRxiv papers of Dutta et al and Tamborrini et al. Now we can actually see the folded-back configuration of myosin crossbridges in relaxed muscle. The current study of Hessel et al adds to the excitement with their novel use of recently developed techniques to answer important questions. They use Harris’s innovative use of SpyCatcher first used in Napierski et al (Circ Res, 2020), to delete the N-terminal domains C1 to C7 within the same preparation. It also builds on the fine work by Hessel et al (PNAS,2022) on the effects of cleaving titin’s I-band region. Here the authors report on the mechanical and structural effects of MyBP-C N-terminal deletion at two sarcomere lengths.

The title of the paper “MyBP-C form C-links ...” gives the impression that structural evidence of C-links will be presented and that the authors are claiming priority for the discovery. In fact, both Tamborrini et al and Huang et al (J. Muscle Res Cell Motil,2023) show cryo-em images of the actual link formed by MyBP-C to actin. On the other hand, in the current paper, C-links are mentioned in the introduction and then never mentioned again. The title needs to be amended to avoid this confusion.

Understanding the role of MyBP-C in striated muscle is obviously important considering the skeletal and

cardiac diseases that occur following mutations in the protein. The current work is an important step towards this goal. The authors have combined the results of their own investigations and with lots of earlier research to build an overall picture. Their main conclusions of the role of MyBP-C are summarised in Fig 3G which shows the many different paths that can be used for fine-tuning muscle contractility.

We are happy to report that we were able to address the reviewer's comments. Below we provide the original decision letter and our replies in a red font. Changes to the text that are reproduced here are indicated by a blue font. Within the manuscript, changes are shown in red. In the rebuttal letter only, we show (name, year) citation style, so readers can quickly identify the citation.

REVIEWER COMMENTS

Reviewer #1 (Remarks to the Author):

The manuscript describes experiments aimed at understanding the role of myosin binding protein-C (MyBP-C) in the contraction of skeletal muscle, and in particular to determine the interactions of MyBP-C with both thin and thick filaments within the myofilament lattice in-situ. The authors developed a SNOOPC2 homozygous mouse line to induce specific cleavage of the N-terminal region of MyBP-C, the C1C7 domains, via TEV protease in demembranated muscle fiber bundles from fast-twitch psoas skeletal muscle. From mechanical experiments on these samples, the authors conclude that MyBP-C cleavage decreases calcium-sensitivity of force and modulates crossbridge cycling kinetics at short sarcomere length. TEV cleavage did not affect passive force. From X-ray diffraction experiments performed in relaxed psoas bundles at different sarcomere lengths, the authors show that cleavage of MyBP-C C1C7 causes expansion of the myofilament lattice and extension of the thin and thick filaments, accompanied by disordering of the myosin heads, suggesting that MyBP-C forms C-links with the actin filament and is important to stabilize myosin OFF state. The idea behind the experiments is logical and clear and the experimental approach should allow to have paired controls before and after TEV cleavage of MyBP-C C1C7 removing some of the variability between samples. However, I have a few questions on the methodology and data interpretation.

We thank the reviewer for their thoughtful review of our manuscript. Our revised changes, outlined below, have significantly improved the quality of our manuscript.

Major points:

In the main text (second paragraph of page 3) the authors state that mechanical data were recorded “in permeabilized fiber bundles before and after cleavage...”; however, in methods, beginning of page 14, they say that “For mechanical measurements in permeabilized psoas from SNOOPC2 mice, psoas myocytes were mechanically homogenized... Psoas cells were attached...” Could the authors clarify what samples are used for the mechanical experiments? Muscle fibers cannot be prepared by homogenization.

We understand the confusion caused by using the term “homogenization” when preparing skeletal fibers. In our case, we used mechanical disruption to prepare fibers for force measurements by using a Polytron homogenizer (pulsed for 2-3 sec on a low setting) to mechanically release individual fibers from larger chunks of muscle followed by detergent-permeabilization as described below. Permeabilized fibers were then attached to a force transducer and positional motor using silicone adhesive. The preparation and attachment method are routinely used for preparation of more fragile cardiac myocytes by our lab and others. An example of a fiber and attachment are shown below and are now included in the data Supplement

(Figure 1). We have also now included a more thorough description of these procedures in the methods section.

Starting from page 8:

“For mechanical measurements in permeabilized psoas from SNOOPC2 mice, muscle fibers were isolated by gentle mechanical disruption (2-3 seconds at a low setting using a Polytron Homogenizer PT1200E, Kinematica, Switzerland) in a skinning solution (5.92 mM Na₂ATP, 6.04 mM MgCl₂, 2 mM EGTA, 139.6 mM KCl, 10 mM imidazole, 0.01% saponin, 1% Triton-X-100®, Halt protease inhibitor cocktail, EDTA-free [78437, Thermo Fisher Scientific, USA]). Following mechanical disruption, fibers tumbled for 30 minutes at 4°C to allow for the removal of sarcoplasmic reticulum, sarcolemma, and any remaining endogenous Ca²⁺, leaving intact the myofibrillar network. Psoas fibers were then thoroughly washed in solution without detergents.”

The authors show that psoas muscle fiber bundles (myocytes?) have a big LDA response, especially at pCa 4.5: the force measured at sarcomere length (SL) 2.7μm is more than double that developed at 2.4μm SL. This is not consistent with the tension-SL relationship of skeletal muscle.

The reviewer is correct that force at 2.4 μm SL is smaller than expected and we appreciate their critical eye. However, the discrepancy is most likely accounted for by compliance inherent in the glue attachment. Consistent with this we found that when fibers were passively stretched to 2.4 and 2.7 μm SL fibers and then maximally activated at pCa 4.5 so that they shortened and developed force at ~2.1 and 2.4 μm SL, respectively (as measured by video recordings). We next conducted an additional series of experiments where fibers were first passively stretched to 3.0 μm SL prior to activation and found that these fibers shortened and developed force at ~2.7 μm, but that there was no significant difference in maximal force generated at 2.7 *versus* 3.0 μm SL (data now included in Supplementary Figure 1). Therefore, the data are consistent with compliance shifting the apparent length-tension curve such that the passive stretch to 2.4 μm SL corresponds to the ascending limb while the passive stretch to 2.7 and 3.0 μm SL corresponds to the plateau. We now include the additional data set at 3.0 μm SL in Figure 1 and Supplementary Figure 1. However, please also note that compliance is expected to be minimal under the conditions used for X-ray experiments because a different attachment method was used to secure the fibers in the beam and fibers did not shorten because X-ray data was collected only under passive (relaxing) conditions and with a different approach.

From updated Fig. 1:

From Extended Data Figure 1:

Results from page 2:

“We used psoas muscle from homozygous SNOOPC2 mice for this evaluation because nearly all fibers are of fast-twitch composition (Hettige et al., 2022). Western blots confirmed successful cleavage of fMyBP-C in permeabilized homozygous SNOOPC2 psoas (Fig. 1e), while not targeting sMyBP-C (Supplementary Fig. 1b) or wildtype fMyBP-C (Fig. 1e). Furthermore, passive tension of fMyBP-C was not affected by TEV_P treatment in fibers from homozygous or wildtype mice in absolute or relative terms (Fig. 1f; Supplementary Fig. 1c). We next evaluated the tension-pCa (-log[Ca²⁺]) relationship in permeabilized fiber bundles before and after cleavage of N-terminal domains of fMyBP-C at shorter (2.4 μm), middle (2.7 μm) and longer (3.0 μm) SLs (Fig. 1g-i; Supplementary Fig. 1c). Cleavage caused a significant rightward shift of the tension-pCa relationship, as quantified by the pCa₅₀ (i.e., the pCa at which active force was half-maximal) at all SLs (Fig. 1g-i). Although pCa₅₀ was altered, LDA was still observed from short to middle and long SLs (i.e., increasing pCa₅₀) in samples both before and after MyBP-C^{C1C7} removal (overlaid in Supplementary Fig. 1d-e). While cleavage impacted submaximal activation levels, it had no significant effect on absolute maximum tension across SLs (Supplementary Fig. 1d-e). Next, we assessed the rates of force redevelopment following a slack/restretch maneuver (k_{tr}), a measure of crossbridge cycling, across a range of calcium concentrations (Fig. 1j-m). At 2.4 and 2.7 μm SL, the k_{tr} vs. force relationships were similar to previous studies (Korte et al., 2003; Song et al., 2021), with k_{tr} increasing with relative force (Fig. 1k-l). Qualitatively, cleaved samples produced a consistent overshoot of the tension near pCa₅₀ before settling to expected values (example in Fig. 1j), which has been reported previously in fast twitch and slow twitch fibers (Robinett et al., 2019). At both short and middle SLs, the loss of fMyBP-C^{C1C7} significantly increased k_{tr} at intermediate but not low or maximal active force levels, while k_{tr} was not significantly different at the longer SL (Fig. 1k-m). Therefore, k_{tr} was less affected by fMyBP-C cleavage at longer vs. shorter SLs. These data are indicative of faster rates of myosin crossbridge cycling after cleavage of fMyBP-C at intermediate calcium activation, consistent with effects observed following loss of cardiac MyBP-C^{C0C7} in cardiomyocytes, and in cardiac and fast skeletal knockout mice (Korte et al., 2003; Song et al., 2021). Detailed statistical information for mechanical datasets is provided in Supplementary Tables 1-8.

Taken together, these data suggest that fMyBP-C^{C1C7} modulates force generation and crossbridge cycling kinetics, specifically at moderate levels of activation (~40-80% of maximal force), in agreement with previous observations in a cardiac MyBP-C TEVp cleavage model, knockout mouse models, and *in vitro* biochemical studies (Geist and Kontogianni-Konstantopoulos, 2016; Harris, 2021; McNamara and Sadayappan, 2018; Song et al., 2021).”

During the X-ray experiments, SL was measured by laser diffraction in bundles of 10-20 fibres. Following from the previous point, what is the error in the SL measurement using laser diffraction?

From laser diffraction, we know that the SL variability can be up to approximately $\pm 0.05 \mu\text{m}$ SL. Because of our fiber bundle prep procedure (we use a suture to wrap knots around the ends of bundles before removed from the whole muscle), our fibers are still in register and stay in register during experiments. We now insert this detail into the methods (see next comment).

Did the authors measure SL before X-ray exposure at each sarcomere length?

SL control is certainly an important consideration. Away from the beamline, we checked with laser diffraction that the mechanics rig used for X-ray experiments stretched samples correctly. During X-ray experiments, the samples were measured via laser diffraction at $2.4 \mu\text{m}$ SL and then mechanically stretched to the next SL. We did not re-check SL during experiments, as the preparations are closed off in a hutch, and the procedure to open and close the door to check every SL would have added hours to already a very tight research schedule at the beamline. As a final level of quality control, during X-ray experiments, there is an inline camera on the preparation that allows for the observation of fibers during experiments, from which we can detect slippage, fiber detachment, or fiber damage.

We now incorporate these details into the methods:

From page 8:

“An inline camera allowed for initial alignment with the X-ray beam and continuous visual quality control of samples during the experiment (e.g. check for fiber slippage and preparation damage). SL was measured via laser diffraction using a 4-mW Helium-Neon laser at $2.4 \mu\text{m}$ SL, with an SL variability of about $\pm 0.05 \mu\text{m}$ SL, based on the width of the 1st-order diffraction. The force baseline was set at slack length. After this initial setup, fiber length changes were accomplished through computer control of the motor, where studies before beamline experiments confirmed appropriate SL change.”

The high spread in d10 (35-42 nm at $2.4 \mu\text{m}$ SL before TEV in Fig.2C) is indicative of a big error in the SL measurement. This might also affect the observed changes in the other X-ray reflections between 2.4 - $2.7 \mu\text{m}$ SL.

As the reviewer notes, we also considered this large spread very carefully over several years of experimentation. As explained above, our preparations have a relatively typical SL variability for these experimental protocols, and we have quality control procedures in place to check that preparations are at the desired SLs. While differences in SL across the sample will certainly contribute to some variability in the lattice spacing, there is also an inherent variability in lattice spacing among sarcomeres at the same length, which may be bigger in permeabilized vs. intact samples. We think that it is likely caused by a variation in the proportion of slow and fast fibers that are within each sample, but this has never been empirically evaluated.

Regardless of the above consideration, it is critical for the reader to understand that although SL may be generally bigger or smaller for each preparation, this will not impact our ability to detect the directionality

of differences between conditions across SLs (e.g., decreasing LS with increasing SL, or increasing SL with MyBP-C cleavage). Each sample was tested at each condition and so we conducted a repeated-measures ANOVA model with a repeated-measures random effect of individual. This means that the basal variability between preparations can be removed from the experimental variability (similar to normalizing to a starting value), and therefore allows for assessment of the directionality of data between conditions.

We now bring this important detail to the readers. The introduction touches on this point.

From page 2:

“This powerful experimental approach allows for the study of changes that occur before and after removal of MyBP-C^{C1C7} *within the same preparation*, removing variation caused by using different samples between conditions and improves statistical power.”

We also provide further context in the results and discussion section when first discussing the X-ray data.

From page 3:

“Generally, structural parameters can be variable between samples, which reduces statistical power to detect differences between conditions. However, our experimental approach of conducting all experimental treatments within one sample allows for the unusual ability to conduct repeated-measures statistical analyses that improved statistical power to assess the directionality of differences between conditions.”

N numbers for mechanical and X-ray data (number of mice and number of fiber bundles/myocytes(?)) are not specified in methods or in the figure legends.

We now include these data in all main text and supplementary figures, and table legends. For example:

From Fig. 1 Legend:

“Experimental dataset derived from 49 fiber bundles from psoas muscles of 10 SNOOPC2 mice (6 female / 4 male).”

Minor points:

The authors say that the data in Fig 1L-M have been normalized by pCa 4.5, however data at pCa 4.5 are not shown in Fig 1G-I.

The updated figures now show the full range of pCa concentrations used.

The methods state that ktr is measured as mono-exponential fit of force redevelopment. Could the authors explain what the dashed lines in Fig 1J-K represent? Please add ticks to the y-axis of panels J-K. The changes in % length change applied during the release-restretch protocol should be shown.

The dashed lines ended up not being helpful and were removed. The other requests from the reviewer were completed.

The fiber bundles were exposed 3 x 1s at three increasing SL, then the protocol was repeated after TEV cleavage (30 min). To estimate the effect of TEV/radiation damage on wildtypes the authors only use the shorter SLs (2.4-2.7 μ m). How can the authors exclude that stretching to 3 μ m might trigger an irreversible change per se?

Limiting radiation damage is a critical point. For control experiments, we only used 2.4-2.7 μ m SL because of time limit considerations at the beamline. However, because of similar concerns in the past, we

previously published a report of repeated measurements using the same X-ray setup with mouse psoas experiments at 2.4 / 2.7 / 3.0 μm SL and found no differences from the first to second stretches (Hessel et al., 2022). We added these details into the methods section.

From page 9:

“To limit X-ray exposure of any one part of the preparation, no part of the sample was exposed more than once. Furthermore, we previously conducted controls on mouse psoas where experiments were run as described here but without TEV_p incubation, and experimental values from the “pre” to “post” trials were consistent (Hessel et al. 2022).”

- Is the X-ray diffraction pattern in Fig. 2A an example from a single 1s exposure, or is it from added psoas bundles? Add exposure time and SL. It is impressive that it was possible to analyze data from each psoas fiber bundle independently, implying that data have good S:N. Could the authors add a supplementary figure showing an example of the 1D profiles (equatorial and meridional) from a single bundle at the three SLs before and after TEV cleavage.

These are indeed from a single exposure on a psoas fiber bundle. We now add the exposure time and SL of the diffraction pattern into the figure legend. We also provide the additional Supplementary figure with 1D plots, as requested (Supplementary Figure 3).

From Figure 2 legend:

“X-ray pattern after 1s exposure, with sample at 2.4 μm SL before treatment. Exemplar 1D profile plots are provided in Supplementary Figure 3.”

- Page 2, 5 lines from bottom of page. “Free-heads” was not defined earlier. Is this really needed here?

The term “free-head” was not needed and was removed.

- Page 3 first line. Could you clarify the calculation used to obtain the number of myosin heads “under the control of MyBP-C”?

For each half-thick filament, there are 9 crowns with binding locations for MyBP-C in the C-zone, as recently confirmed by several CryoEM studies (Dutta et al., 2023; Huang et al., 2023; Tamborrini et al., 2023). 9 crowns x 3 myosin heads per crown x 2 half-thick filaments = 54 total MyBP-C molecules possible. Some animal species / muscles carry 1 or 2 crowns less, which is why we say “up to” 54 total MyBP-C molecules. We now add some further details to our calculations.

From page 2:

“The mechanism(s) by which MyBP-C affects the myosin ON/OFF state are not clear, but the phenomenon is remarkable because there are relatively few MyBP-C (up to ~54 covering 9 crowns per half-thick filament) molecules to regulate the behavior of up to ~300 myosin heads per thick filament.”

- Fig 2A. M2 cluster magnification: the general reader would benefit by identifying with a rectangle the region of the 2D pattern that was magnified.

Agreed and done as requested. We also added dotted lines to track the expansion. Below is the modified panel:

- Legend of Fig.2F-G. Left- and right-handed actin helix are inverted with respect to the cartoon in Fig 2E and the main text.

This error is now fixed.

- Fig 2G. Check ticks of y-axis.

This error is now fixed.

- Page 4 first paragraph. Could the authors discuss the result that C-links might produce radial and longitudinal forces, but have no effect on passive force (Fig 1F)?

Yes, this is an interesting finding that makes us think that several parallel processes are going on, outside of just what fMyBP-C is doing. Furthermore, fMyBP-C could affect these other processes, so it is hard to separate them. Since the original submission, we have refined our ideas on this topic, and now re-worked this section. We provide the most relevant parts of the section below.

From page 3:

“We first considered a long-debated question in muscle physiology: are there load-bearing MyBP-C “C-links” between thick and thin filaments? A series of recent imaging studies in relaxed cardiac sarcomeres provide compelling evidence that MyBP-C can interact with thin filaments (Huang et al., 2023; Tamborrini et al., 2023), but the strength of these interactions is unclear. The interdigitating thick and thin filaments of the sarcomere form a hexagonal lattice, where 6 thin filaments surround a thick filament (Fig. 2b). Mechanically speaking, in passive muscle, any protein linking the thick to thin filaments should pull on the thin filaments as SL increases, which would be expected to produce both radial forces that pull the filaments together and longitudinal forces that elongate the thin filaments. Therefore, we used our X-ray dataset to assess lattice spacing and thin filament length before and after fMyBP-C^{C1C7} removal...

...From our findings, it would be logical to postulate that cleaving fMyBP-C^{C1C7} terminates the thick-thin filament bridge and removes any MyBP-C-based pulling forces placed on it, which would both increase lattice spacing and shorten the thin filament. However, based on the statistically insignificant different passive tension drop after fMyBP-C^{C1C7} removal (Fig. 1f), the magnitude of MyBP-C-based pulling forces on the lattice are most likely small and may not be able to account for these structural changes alone.

Furthermore, while fMyBP-C^{C1C7} removal decreased overall thin filament length, it did not eliminate thin filament elongation during passive sarcomere stretch. Therefore, it seems that parallel processes are occurring in the sarcomere. We posit two possibilities that are not necessarily mutually exclusive. First, the uncleavable slow-MyBP-C^{C1C7}s, although few in psoas muscle, could nevertheless form C-links and contribute to stretch-dependent thin filament extension. If slow-MyBP-C^{C1C7} contributes disproportionate forces on the lattice, then this could explain why passive tension did not change much after fMyBP-C^{C1C7} removal, and why thin-filament length still extended with sarcomere stretch. The second relies on the known property that low-level, force-producing crossbridges occur in passive muscle (Donaldson et al., 2012; Selby et al., 2011) and so can contribute to thin filament stretch. It is possible that fMyBP-C^{C1C7} removal alters this behavior, as is supported by other studies, for example (Song et al., 2021), and this study (also see structural data on myosin heads detailed below), which show both altered crossbridge kinetics and loss of myosin head suppression when MyBP-C is altered. The potential interplay between these scenarios, most likely running in parallel and somewhat interconnected, makes it difficult to study each individually but could potentially be separated out using computational modeling and is worth exploring (Prodanovic et al., 2023).”

- Fig. 3A should show changes in the M2 cluster within the 2D pattern: could the 2D magnification have the same orientation as in Fig.2A? Is the equator at the top in Fig 2A and at the bottom in Fig. 3A? In Fig 2A the magnification at the level of the M2 cluster includes the second order of the troponin reflection. Was the troponin reflection included within the integration limits used to calculate the intensity of the M2 cluster? Could the authors provide the integration limits they have used (not reported in methods)?

We originally showed non-M2 cluster reflections in Fig 3A and Fig. 2A, but we did not integrate these. We now only show the M2 clusters in the same orientation for Fig. 3A and 2A (equator above). We further now report the integration limits for the M2 cluster and all others in the methods.

From page 9:

“The meridian intensities along the meridional axis ($I_{M2_Cluster}$, S_{M3} , S_{T3} , S_{M6}) were first calculated by integrating in the reciprocal radial range $\sim 0 \leq R \leq 0.032 \text{ nm}^{-1}$ for M3, M6, and T3 reflections, and $\sim 0.013 \leq R \leq 0.053 \text{ nm}^{-1}$ for the A6 and A7 reflection, where R denotes the radial coordinate in reciprocal space (Wakabayashi et al., 1994). Next, integration limits along the median axis were as follows: M2 cluster $0.04 \leq R \leq 0.05 \text{ nm}^{-1}$; M3, $0.066 \leq R \leq 0.074 \text{ nm}^{-1}$; M6, $0.136 \leq R \leq 0.144 \text{ nm}^{-1}$; T3, $0.078 \leq R \leq 0.080 \text{ nm}^{-1}$; A6, $0.166 \leq R \leq 0.175 \text{ nm}^{-1}$; A7, $0.192 \leq R \leq 0.204 \text{ nm}^{-1}$. From these, the spacing and intensity of each reflection are calculated automatically. The S_{A6} and S_{A7} report on the left- and right-handed actin helical structures within the thin filament were used here to calculate the axial spacing of the actin monomers (S_{gActin} ; Equation 1), where S_{gActin} can be used as a measure of thin filament extension (Egelman et al., 1982; Wakabayashi et al., 1994).”

$$gActin \text{ spacing} = \frac{1}{\left(\frac{1}{A6 \text{ Spacing}}\right) + \left(\frac{1}{A7 \text{ Spacing}}\right)} \quad (1)$$

- Page 5, second paragraph. “..., with increasing I11/I10 indicating that more myosin heads are associated with the thin filaments”. This sentence should be rephrased as the experiments were performed in relaxing conditions and the changes in I11/I10 are small compared to the changes observed between resting-active muscles (see your cited literature).

We now clarify that the increase in I11/I10 in relaxing conditions is indicative of an increase in the propensity of myosin heads to form crossbridges upon activation.

From page 5:

“It provides a measure of the transfer of mass (i.e., myosin heads) from the thick to the thin filaments, with increasing $I_{1,1}/I_{1,0}$ indicating that more myosin heads are in a favorable orientation to associated with the thin filaments upon contraction (Ma et al., 2018).”

- Page 5, second paragraph, line 5-6: “myosin head configuration can be evaluated via the spacing of the M3 myosin meridional reflection (SM3)”: this is an extrapolation. SM3, as you say later, is a measurement of the axial spacing of the myosin heads. To measure changes in heads conformation the authors might need to also measure changes in the intensity of the M3 reflection and in its fine structure, and a model.

The reviewer makes a valid point that several pieces of evidence are needed to solidify the story of myosin head position, including intensity measures, as well as MLL1 and MLL4 reflections – other useful indications of the radial distance of myosin heads away from the thick filament backbone. These are not resolved enough in permeabilized preps to collect. However, we do have other markers that help strengthen our argument, such as the equatorial intensity ratio. Generally, intact preparations would provide more robust details, but this is not yet possible in our preparations, as TEV protease can only pass through permeabilized fiber membranes. We hope to make a technical breakthrough in that regard soon. These technological limitations are important to acknowledge and so we add a section about it called “Limitations and outlook”.

From page 6:

“Known limitations of the TEV protease procedure are that membrane permeabilization can expand the sarcomere lattice while experiments at 25°C naturally reduce myosin head order and partially shift heads towards the ON state. Nevertheless, there is no a priori reason to expect that the directionality of the effects of MyBP-C cleavage on the myofilaments would be different from in vivo conditions. Therefore, identifying a strategy to rapidly cleave fMyBP-C^{C1C7} using intact preparations is an immediate next goal, which would then also provide more detailed reflection information, such as myosin layer lines 1 and 4, and higher order reflections that would further illuminate the orientation of myosin heads within the sarcomeres. We are therefore excited that technology is now poised to explore MyBP-C function in otherwise healthy, living sarcomeres and is a promising new approach within muscle physiology that will only improve going forward.”

- Page 5, second paragraph, line 10, and 2 lines from bottom: “behaved as expected” please rephrase, e.g. in agreement with previously published work.

Done as requested.

- Model of thick filament activation, page 6 and Fig.3 G. The proposed model is plausible but will need further validation. As the authors include in their model the information from the recent cryo-EM structures (Dutta et al., Tamborini et al.), could they discuss if and how the presence of the TEV protease cleavage site and SnoopTag site between C7-C8 of MyBP-C might perturb the interactions with Crown 1 or Crown 2 tails per se?

This is a great point. The location of the MyBP-C cleavage site between domains C7 and C8 and the binding sites of C8, C9, and C10 to the thick filament identified by cryo-EM are not expected to overlap based on their locations from the recent structural studies (Dutta et al., 2023; Huang et al., 2023; Tamborini et al., 2023).

From page 2:

“The short SnoopTag insertion occurs upstream of C8 and so should not disrupt the proposed OFF-state stabilizing role of the Cr1 and Cr3 heads, as recently described (Dutta et al., 2023; Huang et al., 2023; Tamborrini et al., 2023).”

- Related to the previous point, could the authors run a non-paired t-test to compare the data from X-ray signals obtained before TEV in control wildtype and homozygous SNOOPC2 psoas bundles (e.g. data in Extended data Fig.2 with data in Fig. 2 and 3)?

Yes, and we also took the analysis a step further. We now include an analysis between the SNOOPC2 before and control data for 2.4 and 2.7 μm SL. Because we have repeated measures within each sample across sarcomere lengths but not across genotypes, we conducted an ANOVA with fixed effects sarcomere length, genotype, and length x genotype, and a random effect of individual nested within genotype. The analysis suggests parameters are generally similar between SNOOPC2 and wildtype mouse, except for Sig.D, which is larger in controls, suggesting that there is more lattice spacing heterogeneity in controls vs. SNOOPC2 samples. We are not sure if this is meaningful in permeabilized samples, as the length of permeabilization changes the sig. D, and as a logistical need, control samples were in permeabilization solution for ~1 additional week than SNOOPC2 samples.

The total analysis is now included in Supplementary Table 8. We also now refer to it in the main text as well.

From page 3:

“Finally, a statistical analysis between control and SNOOPC2 datasets did not reveal any major structural differences, with further statistical details available in Supplementary Table 12.”

- Page 12. Legend of Fig.3D: rephrase. M3 spacing SM3 measures the axial spacing of the myosin heads. Fig 3E legend: M6 spacing SM6 measures the spacing of the thick filament backbone.

Done as suggested.

- Page 13. Western blot, 2nd line. “left ventricle tissue was homogenized..” why the authors used left ventricle tissue when the rest of the experiments were performed on psoas muscle?

We apologize for this typo. We now correct this to say psoas muscle.

- Page 14. Small-angle X-ray diffraction experiments. 6th line: “For TC experiments...” please expand TC.

TC was a typo and now removed.

Bottom line: could you report the measured diameter of fiber bundles?

Yes. The fiber diameter was 0.49 ± 0.02 mm. We placed this detail into the methods.

- Page 15. X-ray image analysis. 9 lines from bottom. “Meridional (IM2cluster, SM3, IM3,..” IM3 is not shown. Could the authors add integration limits for the X-ray reflections analyzed (equatorial and meridional)? Could the authors also report integration limits for the actin layer lines and the formula used to calculate Sgactin?

Completed as requested.

On page 9:

“The equator of the X-ray pattern was determined by integrating the intensities along the equatorial axis ($\pm 0.005 \text{ nm}^{-1}$ on either side of the equatorial axis), followed by 1,0 and 1,1 fitting Gaussian curves to the 1,0 and 1,1 reflection, respectively. Spacing and intensities were calculated from maximum Gaussian peak and area under the Gaussian curve, respectively. The meridian intensities along the meridional axis ($I_{M2_Cluster}$, S_{M3} , S_{T3} , S_{M6}) were first calculated by integrating in the reciprocal radial range $\sim 0 \leq R \leq 0.032 \text{ nm}^{-1}$ for M3, M6, and T3 reflections, and $\sim 0.013 \leq R \leq 0.053 \text{ nm}^{-1}$ for the A6 and A7 reflection, where R denotes the radial coordinate in reciprocal space (Wakabayashi et al., 1994). Next, integration limits along the median axis were as follows: M2 cluster $0.04 \leq R \leq 0.05 \text{ nm}^{-1}$; M3, $0.066 \leq R \leq 0.074 \text{ nm}^{-1}$; M6, $0.136 \leq R \leq 0.144 \text{ nm}^{-1}$; T3, $0.078 \leq R \leq 0.080 \text{ nm}^{-1}$; A6, $0.166 \leq R \leq 0.175 \text{ nm}^{-1}$; A7, $0.192 \leq R \leq 0.204 \text{ nm}^{-1}$. From these, the spacing and intensity of each reflection are calculated automatically.”

• Extended data tables: Could you explain all used parameters in the legend? (e.g. does Tn-pCa stands for tension-pCa ?).

Yes, but this phrase was not needed, so we now use Tension (mN/mm^2) and use another column to indicate pCa. We made these changes throughout all Tables.

• Ext data table 2: What is the absolute value of ktr?

We now add another table with the absolute ktr values (Supplementary Table 4).

Ext data table 3-4: What are the units of measurement of absolute force?

We used tension (mN/mm^2). These are now included.

Ext data table 6: Are the mean numbers associated to IM2cluster normalized values? If yes, what would correspond to IM2cluster=1?

Data is normalized to the total background subtracted for each image within the MuscleX quadrant folding routine. We now provide those details in the methods.

From page 9:

“The Quadrant Folding routine was used to improve the signal-to-noise by adding together the four equatorial-meridional quadrants, which each provide the same information (Friedel’s Law), followed by the built-in circularly symmetric background subtraction protocol. Reflection intensity values were divided by the total subtracted background to normalize all images.”

Reviewer #2 (Remarks to the Author):

This is an exciting time in muscle research with the recent BioRxiv papers of Dutta et al and Tamborrini et al. Now we can actually see the folded-back configuration of myosin crossbridges in relaxed muscle. The current study of Hessel et al adds to the excitement with their novel use of recently developed techniques to answer important questions. They use Harris's innovative use of SpyCatcher first used in Napierski et al (Circ Res, 2020), to delete the N-terminal domains C1 to C7 within the same preparation. It also builds on the fine work by Hessel et al (PNAS,2022) on the effects of cleaving titin's I-band region. Here the authors report on the mechanical and structural effects of MyBP-C N-terminal deletion at two sarcomere lengths.

We thank the reviewer for their enthusiasm about this paper.

The title of the paper "MyBP-C form C-links ..." gives the impression that structural evidence of C-links will be presented and that the authors are claiming priority for the discovery. In fact, both Tamborrini et al and Huang et al (J. Muscle Res Cell Motil,2023) show cryo-em images of the actual link formed by MyBP-C to actin. On the other hand, in the current paper, C-links are mentioned in the introduction and then never mentioned again. The title needs to be amended to avoid this confusion.

We understand this criticism and now change the title to:

"Myosin-binding protein C regulates the sarcomere lattice and stabilizes the OFF states of myosin heads"

Understanding the role of MyBP-C in striated muscle is obviously important considering the skeletal and cardiac diseases that occur following mutations in the protein. The current work is an important step towards this goal. The authors have combined the results of their own investigations and with lots of earlier research to build an overall picture. Their main conclusions of the role of MyBP-C are summarised in Fig 3G which shows the many different paths that can be used for fine-tuning muscle contractility.

We again thank the reviewer for their review of our manuscript.

References

- Donaldson, C., Palmer, B. M., Zile, M., Maughan, D. W., Ikonomidis, J. S., Granzier, H., Meyer, M., VanBuren, P. and LeWinter, M. M.** (2012). Myosin cross-bridge dynamics in patients with hypertension and concentric left ventricular remodeling. *Circ. Heart Fail.* **5**, 803–811.
- Dutta, D., Nguyen, V., Campbell, K. S., Padrón, R. and Craig, R.** (2023). Cryo-EM structure of the human cardiac myosin filament. *Nature*.
- Egelman, E. H., Francis, N. and DeRosier, D. J.** (1982). F-actin is a helix with a random variable twist. *Nature* **298**, 131–135.
- Geist, J. and Kontrogianni-Konstantopoulos, A.** (2016). MYBPC1, an emerging myopathic gene: what we know and what we need to learn. *Front. Physiol.* **7**, 410.
- Harris, S. P.** (2021). Making waves: A proposed new role for myosin-binding protein C in regulating oscillatory contractions in vertebrate striated muscle. *J. Gen. Physiol.* **153**,.

- Hessel, A. L., Ma, W., Mazara, N., Rice, P. E., Nissen, D., Gong, H., Kuehn, M., Irving, T. and Linke, W. A.** (2022). Titin force in muscle cells alters lattice order, thick and thin filament protein formation. *Proc Natl Acad Sci USA* **119**, e2209441119.
- Hettige, P., Tahir, U., Nishikawa, K. C. and Gage, M. J.** (2022). Transcriptomic profiles of muscular dystrophy with myositis (mdm) in extensor digitorum longus, psoas, and soleus muscles from mice. *BMC Genomics* **23**, 657.
- Huang, X., Torre, I., Chiappi, M., Yin, Z., Vydyanath, A., Cao, S., Raschdorf, O., Beeby, M., Quigley, B., de Tombe, P. P., et al.** (2023). Cryo-electron tomography of intact cardiac muscle reveals myosin binding protein-C linking myosin and actin filaments. *J. Muscle Res. Cell Motil.*
- Korte, F. S., McDonald, K. S., Harris, S. P. and Moss, R. L.** (2003). Loaded shortening, power output, and rate of force redevelopment are increased with knockout of cardiac myosin binding protein-C. *Circ. Res.* **93**, 752–758.
- Ma, W., Gong, H. and Irving, T.** (2018). Myosin head configurations in resting and contracting murine skeletal muscle. *Int. J. Mol. Sci.* **19**,.
- McNamara, J. W. and Sadayappan, S.** (2018). Skeletal myosin binding protein-C: An increasingly important regulator of striated muscle physiology. *Arch. Biochem. Biophys.* **660**, 121–128.
- Prodanovic, M., Wang, Y., Mijailovich, S. M. and Irving, T.** (2023). Using Multiscale Simulations as a Tool to Interpret Equatorial X-ray Fiber Diffraction Patterns from Skeletal Muscle. *Int. J. Mol. Sci.* **24**,.
- Robinet, J. C., Hanft, L. M., Geist, J., Kontrogianni-Konstantopoulos, A. and McDonald, K. S.** (2019). Regulation of myofilament force and loaded shortening by skeletal myosin binding protein C. *J. Gen. Physiol.* **151**, 645–659.
- Selby, D. E., Palmer, B. M., LeWinter, M. M. and Meyer, M.** (2011). Tachycardia-induced diastolic dysfunction and resting tone in myocardium from patients with a normal ejection fraction. *J. Am. Coll. Cardiol.* **58**, 147–154.
- Song, T., McNamara, J. W., Ma, W., Landim-Vieira, M., Lee, K. H., Martin, L. A., Heiny, J. A., Lorenz, J. N., Craig, R., Pinto, J. R., et al.** (2021). Fast skeletal myosin-binding protein-C regulates fast skeletal muscle contraction. *Proc Natl Acad Sci USA* **118**,.
- Tamborrini, D., Wang, Z., Wagner, T., Tacke, S., Stabrin, M., Grange, M., Kho, A. L., Rees, M., Bennett, P., Gautel, M., et al.** (2023). Structure of the native myosin filament in the relaxed cardiac sarcomere. *Nature* **623**, 863–871.
- Wakabayashi, K., Sugimoto, Y., Tanaka, H., Ueno, Y., Takezawa, Y. and Amemiya, Y.** (1994). X-ray diffraction evidence for the extensibility of actin and myosin filaments during muscle contraction. *Biophys. J.* **67**, 2422–2435.

REVIEWER COMMENTS

Reviewer #1 (Remarks to the Author):

I thank the authors for their clear replies to my questions and additional explanations/figures added to the text.

As minor points I would like them to clarify:

- Page 3, 6th line, please clarify that you do not have LDA, but that you have compliance in the glue attachments and therefore active SL is about 2.1/2.4/2.7 μm . I agree that this should not affect your X-ray measurements as those are collected in passive conditions.
- Extended data figure/Supplementary Figure 1d: this graph appears now different from the one before normalisation in Fig 1f, where at 3 μm passive force is higher after TEV. Did you normalised by the passive force at 3 μm before or after TEV? Could you add the point at $\gamma=1$ so that we can understand this more easily?
- “The fiber diameter was $0.49 \pm 0.02 \text{ mm}$ ”. Is this the diameter of the whole bundle?
- Is σ_D expressed in nm^{-1} , nm or pixels?

I believe that they addressed all of my other concerns.

Reviewer #2 (Remarks to the Author):

I am happy that the authors have addressed satisfactorily the issues raised in my review of the original manuscript.

Below we provide the original decision letter and our replies in a red font. Changes to the text that are reproduced here are indicated by a blue font. Within the marked-up manuscript, changes are shown in red.

REVIEWER COMMENTS

Reviewer #1 (Remarks to the Author):

I thank the authors for their clear replies to my questions and additional explanations/figures added to the text.

As minor points I would like them to clarify:

- Page 3, 6th line, please clarify that you do not have LDA, but that you have compliance in the glue attachments and therefore active SL is about 2.1/2.4/2.7 μm . I agree that this should not affect your X-ray measurements as those are collected in passive conditions.

This clarification was added to the text.

From main text (Page 3):

“However, due to compliance resulting from attaching the cell with silicone rubber, at maximal activation the SL shortened to 2.1 μm (shorter), 2.4 μm (middle), and 2.7 μm (longer).”

- Extended data figure/Supplementary Figure 1d: this graph appears now different from the one before normalisation in Fig 1f, where at 3 μm passive force is higher after TEV. Did you normalised by the passive force at 3 μm before or after TEV? Could you add the point at y=1 so that we can understand this more easily?

The y-axis on the supplementary figure 1d was mislabeled and has now been fixed to show that the 3 μm points at y=1. We thank the reviewer for recognizing this error. For these measurements, all Pre-TEVp treatment measurements were normalized to the passive force at 3 μm before treatment, and all post-treatment measurements were normalized to passive force at 3 μm after treatment. This clarification was also added into the methods section.

From main text (page 8):

“Passive force values were measured in a range of SL (2.2, 2.4, 2.6, 2.8, and 3.0 μm) and plotted as absolute values or normalized to the maximal passive force measured at SL 3.0 μm within the treatment condition.”

- “The fiber diameter was 0.49 ± 0.02 mm”. Is this the diameter of the whole bundle?

This is the fiber bundle. We now say, “the fiber bundle diameter” in the two places this is referred to in the main text (pages 7 and 9).

- Is σD expressed in nm^{-1} , nm or pixels?

σD is expressed as nm^{-1} . We did indicate this in the Fig. 2 axis, and now further include this detail upon the first mention of σD in the main text (page 4).

I believe that they addressed all of my other concerns.

Thank you.

Reviewer #2 (Remarks to the Author):

I am happy that the authors have addressed satisfactorily the issues raised in my review of the original manuscript.

Thank you.

REVIEWERS' COMMENTS

Reviewer #1 (Remarks to the Author):

I thank the authors for replying to my remaining questions. I believe that they addressed all of my points.
Thank you.